# Preliminary investigation of the relationship between differential phase shift and path-integrated attenuation at X-band in an Alpine environment

5  Guy Delrieu[1], Anil Kumar Khanal[1], Nan Yu[2], Frédéric Cazenave[1], Brice Boudevillain[1], and Nicolas Gaussiat[2]

1 Institute for Geosciences and Environmental research (IGE), UMR 5001 (Université Grenoble Alpes, CNRS, IRD), Grenoble, France

Centre de Météorologie Radar, Direction des Systèmes d'Observation, Météo France, Toulouse, France

*Correspondence to*: guy.delrieu@univ-grenoble-alpes.fr

**Abstract.** The RadAlp experiment aims at developing advanced methods for rainfall and snowfall estimation using weather radar remote sensing techniques in high mountain regions for improved water resource assessment and hydrological risk mitigation. A unique observation system has been deployed since 2016 in the Grenoble region, France. It is composed of an
X-band radar operated by Météo-France on top of the Mt Moucherotte (1901 m asl; MOUC radar hereinafter). In the Grenoble valley (220 m asl), we operate a research X-band radar called XPORT and *in situ* sensors (weather station, rain gauge, disdrometer). We present in this article a methodology for studying the relationship between the differential phase shift due to propagation in precipitation ($\Phi_{dp}$) and path-integrated attenuation ($PIA$) at X-Band. This relationship is critical for quantitative precipitation estimation (QPE) based on polarimetry due to severe attenuation effects in rain at the considered
frequency. Furthermore, this relationship is still poorly documented in the melting layer (ML) due to the complexity of the hydrometeors' distributions in terms of size, shape and density. The available observation system offers promising features to improve this understanding and to subsequently better process the radar observations in the ML. We use the Mountain Reference Technique for direct PIA estimations associated with the decrease of returns from mountain targets during precipitation events. The polarimetric PIA estimations are based on the regularization of the profiles of the total differential
phase shift ($\Psi_{dp}$) from which the specific differential phase shift on propagation ($K_{dp}$) profiles are derived. This is followed by the application of relationships between the specific attenuation ($k$) and the specific differential phase shift. Such $k - K_{dp}$ relationships are estimated for rain by using available drop size distribution (DSD) measurements available at ground level. Two sets of precipitation events are considered in this preliminary study: (i) nine convective cases with high rain rates which allows us to study the $\phi_{dp} - PIA$ relationship in rain; (ii) a stratiform case with moderate rainrates, for which the melting layer
(ML) rose up from about 1000 m asl up to 2500 m asl, where we were able to perform a horizontal scanning of the ML with the MOUC radar and a detailed analysis of the $\phi_{dp} - PIA$ relationship in the various layers of the ML. A common methodology was developed for the two configurations with some specific parameterizations. The various sources of error

affecting the two PIA estimators are discussed: stability of the dry-weather mountain reference targets, radome attenuation, noise of the total differential phase shift profiles, contamination due to the differential phase shift on backscatter, relevance of the $k - K_{dp}$ relationship derived from DSD measurements, etc. In the end, the rain case study indicates that the relationship between MRT-derived PIAs and polarimetry-derived PIAs presents an overall coherence but quite a considerable dispersion (explained variance of 0.77). Interestingly, the non-linear $k - K_{dp}$ relationship derived from independent DSD measurements yields almost unbiased PIA estimates. For the stratiform case, clear signatures of the MRT-derived PIA, the corresponding $\phi_{dp}$ value and their ratio are evidenced within the ML. In particular, the averaged $PIA/\phi_{dp}$ ratio, a proxy for the slope of a linear $k - K_{dp}$ relationship in the ML, peaks at the level of the co-polar correlation coefficient ($\rho_{hv}$) peak, just below the reflectivity peak, with a value of about 0.42 dB degree$^{-1}$. Its value in rain below the ML is 0.33 dB degree$^{-1}$, in rather good agreement with the slope of the linear $k - K_{dp}$ relationship derived from DSD measurements at ground level. The $PIA/\phi_{dp}$ ratio remains quite high in the upper part of the ML, between 0.32 and 0.38 dB degree$^{-1}$, before tending towards 0 above the ML.

## 1 Introduction

Estimation of atmospheric precipitation (solid / liquid) is important in a high mountain region such as the Alps for the assessment and management of water and snow resources for drinking water, hydro-power production, agriculture and tourism, characterized by high seasonal variability. One of the most critical application concerns the prediction of natural hazards associated with intense precipitation and melting of snowpacks, i.e. inundations, floods, flash floods and gravitational movements, which requires a high-resolution observation: spatial resolution $\leq 1km^2$ and temporal resolution $\leq 1hr$. While this can hardly be achieved over extended areas with traditional *in-situ* raingauge networks, the use of radar remote sensing has a high potential that needs to be exploited but also a number of limitations that need to be surpassed. Quantitative Precipitation Estimation (QPE) with radar remote sensing in a complex terrain such as the Alps is made challenging by the topography and the space-time structure and dynamics of precipitation systems. Radar coverage of the mountain regions brings the following dilemma. On the one hand, installing a radar at the top of a mountain allows a 360° panoramic view and therefore the ability to detect precipitation systems over a long range at the regional scale. This is particularly relevant for localized and heavy convective systems in warm seasons. But the precipitation is likely to undergo significant change in between detection and arrival at ground level, including a phase change when the 0°C isotherm is located at the level of or lower than the radar beam altitude. Such situations are likely to be frequent during cold periods, with a strong impact on QPE quality at ground level. On the other hand, installing a radar at the bottom of the valley provides high resolution and quality data required for vulnerable and densely populated Alpine valleys, but the QPEs are limited at the latter due to beam blockage by surrounding mountains.

In Europe, MeteoSwiss has the longest-standing experience in operating radars in mountainous regions. The Swiss C-band radar network in the Alps (Joss and Lee, 1995; Germann et al. 2006) is one of the highest in the world and is coping with the

associated altitude dilemma by using a large number of PPI scans (including negative elevation ones) aimed at determining
high resolution vertical profiles of reflectivity. Sophisticated radar-raingauge merging techniques and echo tracking techniques, as well as numerical prediction models outputs (Sideris et al. 2014; Foresti et al. 2018) are implemented to better understand and quantify the complexity of precipitation distribution in such a rugged environment. More recently, Météo-France has chosen to complement the coverage of its operational radar network ARAMIS (for Application Radar à la Météorologie Infra-Synoptique) in the Alps by means of X-Band polarimetric radars. A first set of three radars was installed in Southern Alps within the RHyTMME project (Risques Hydrométéorologiques en Territoires de Montagnes et Méditerranéens) in the period 2008-2013 at Montagne de Maurel (1770 m above sea level, asl), Mont Colombis (1740 m asl) and Vars Mayt (2400 m asl) (Westrelin et al. 2012). This effort has been continued in 2014-2015 with the installation of an additional X-band radar system (MOUC radar, hereinafter) on top of the Mount Moucherotte (1920 m) that dominates the valley of Grenoble, the biggest city in the French Alps with about 500,000 inhabitants. The choice of the X-Band frequency is challenging due to its sensitivity to attenuation (e.g. Delrieu et al. 2000). In the past, the IGE radar team has proposed the so-called Mountain Reference Technique (MRT) (Delrieu et al. 1997; Serrar et al. 2000; Bouilloud et al. 2009) to take advantage of this drawback for both correcting the gate to gate attenuation and performing a self-calibration of the radar. The idea was to estimate path-integrated attenuations (PIA) in some specific directions from the decrease of mountain returns during rainy periods. Such PIA estimates were then used as constraints for backward or forward attenuation correction algorithms (Marzoug and Amayenc 1994) with optimization of an effective radar calibration error, given a drop size distribution (DSD) parameterization. The development of polarimetric radar techniques (e.g. Bringi and Chandrasekar 2001; Ryzhkov et al. 2005) has allowed a scientific breakthrough for quantitative precipitation estimation (QPE) at X-band by exploiting the relationship which exists between the specific differential phase shift on propagation ($K_{dp}$, in ° km$^{-1}$) and the specific attenuation $k$ (dB km$^{-1}$). As with the MRT, the differential propagation phase $\Phi_{dp}(r_2) - \Phi_{dp}(r_1)$ over a given path $(r_1, r_2)$ can be used to estimate $PIA(r_1, r_2)$, which can constrain a backward attenuation correction algorithm and allow a self-calibration of the radar and/or an adjustment of the DSD parameterization (Testud et al. 2000; Ryzhkov et al. 2014). Two major advantages of the polarimetric technique over the MRT can be formulated: (1) the availability of PIA constraints for any direction with significant precipitation and (2) the subsequent possibility to use a backward attenuation correction algorithm, which is known to be stable while the forward formulation is inherently unstable. Accounting for their respective potential in different rain regimes (moderate to heavy), some combined algorithms making use of various polarimetric observables (reflectivity, differential reflectivity and specific differential phase shift on propagation) have also been proposed for the X-Band frequency (e.g. Matrosov and Clark, 2002; Matrosov et al. 2005; Koffi et al. 2014). Although the polarimetric QPE methodology is now quite well established and validated for rainy precipitation (Matrosov et al., 2005; Anagnostou et al. 2004; Diss et al. 2009), Yu et al. (2018) have shown in their first performance assessment of the RHyTMME radar network, the limitations associated with the use of polarimetric X-band radars in mountainous regions and pointed out (i) the need to better understand and quantify attenuation effects in the melting layer (ML), (ii) the importance of non-uniform beam filling (NUBF) effects at medium to long ranges in such a high-mountain context, as well as (iii) the stronger impact of radome attenuation at X-band compared to

S- or C-Band. Yu et al. (2018) had also a first attempt at studying the relationship between the specific differential phase shift on propagation and the specific attenuation in the melting later by using the collocated measurements of two X band radars situated one well below and the other one well above the 0°C isotherm and by considering the attenuation uniform within the ML.

Since 2016, we have the opportunity to operate a research X-Band polarimetric radar system (XPORT radar hereinafter) at IGE at the bottom of the Grenoble valley. This unique facility, consisting of two radar systems 11 km apart operating on an altitudinal gradient of about 1700 m, should enable us to make progress on how to deal with the altitude dilemma and with potential / issues associated with the choice of the X-band operating frequency. Following a first article based on the RadAlp experiment about the characterization of the melting layer (Khanal et al. 2019), we concentrate hereinafter on the relationship between total differential phase shift ($\phi_{dp}$) derived from polarimetry and PIA derived from the MRT. In section 2, we present the observation system available, as well as contrasted rainy events considered in this study: (i) a set of nine convective events with high rain rates, for which the melting layer was well above the detection domain of the XPORT radar, allows us to study the $\phi_{dp} - PIA$ relationship in rain; (ii) a stratiform case with moderate rainrates, for which the melting layer rose up from about 1000 m asl up to 2500 m asl, allows us to perform a horizontal scanning of the ML with the MOUC radar and a preliminary analysis of the $\phi_{dp} - PIA$ relationship in the various layers of the ML. We present and illustrate in section 3 the methodology used for the PIA and $\phi_{dp}$ estimation. We also investigate the relationship between the specific differential phase shift on propagation ($K_{dp}$) and the specific attenuation ($k$) thanks to drop size distribution (DSD) measurements collected in the Grenoble valley during the two sets of events. The results concerning the $\phi_{dp} - PIA$ relationship in rain and in the ML are presented and discussed in section 4, while conclusions and perspectives are drawn in section 5.

## 2. Observation system and datasets

### 2.1. Observation system

Grenoble is a Y-shaped alluvial valley in the French Alps with a mean altitude of about 220 m asl surrounded by three mountain ranges: Chartreuse (culminating at 2083 m asl) to the north, Belledonne (2977 m) to the south-east and Vercors (2307 m) to the west. Figure 1 shows the topography of the area as well as the positions of the Météo-France radar system on top of the Mt Moucherotte and the IGE experimental site at the bottom of the valley.

*Figure 1 here*

Among other devices, the IGE experimental site includes: (i) the IGE XPORT research radar (Koffi et al. 2014); see Table 1 for the list of its main parameters; (ii) one micro-rain radar (MRR, not used in the current study), (iii) one meteorological station including pressure, temperature, humidity, wind probes and several raingauges, (iv) one PARSIVEL2 disdrometer. The characteristics of the MOUC radar are listed in Table 1. XPORT radar was built in the laboratory in the 2000s. It was operated

during more than 10 years in Western Africa within the AMMA and Megha Tropiques Cal-Val campaigns. Since its return in France in 2016, a maintenance and updating program is underway to improve its functionalities, notably with respect to the
real time data processing and the antenna control program. One noticeable feature for XPORT radar is the range bin size of 34.2 m (corresponding actually to an over-sampling since, for a pulse width of 1 μs, the theoretical bin size is 150 m) which is an interesting figure for the close range and volumetric measurements considered in this study. Note that while the MOUC radar is operated 24 hours a day and its data integrated in the Météo France mosaic radar products, the XPORT radar is operated on alerts only for significant precipitation events.

**2.2 Dataset**

Table 2 shows the main characteristics of nine convective events considered for the study of the $\phi_{dp} - PIA$ relationship in rain, by using the XPORT radar data. A stratiform event, which occurred on January 3-4, 2018, is also considered for a preliminary study of the $\phi_{dp} - PIA$ relationship in the ML, with both the MOUC and the XPORT radar data. Figure 2 presents time series of one of the most intense convective event (July 21, 2017) and the stratiform event. In both cases, the total rain
amount observed at the IGE site was about 35 mm, but in 3 hours with two peak rainrates of about 40 mm h$^{-1}$ for the July 21, 2017 convective event while the January 3-4, 2018 stratiform event lasted more than 12 hours with an average rainrate of about 3 mm h$^{-1}$. The two events also differ by their vertical structure. The bottom graphs of Fig. 2 display the time series of the altitudes of the tops, peaks and bottoms of the horizontal reflectivity (Zh) and co-polar correlation coefficient ($\rho_{hv}$) signatures of the ML, obtained with the automatic detection algorithm described in Khanal et al. (2019). The quasi-vertical profiles (QVP,
Ryzhkov et al. 2016) derived from the XPORT 25°-PPIs are considered in the ML detection. For the convective case, the ML extends from 3000 up to 4000 m asl and more, i.e. well above the altitudes of the two radars. Table 2 indicates this is also the case for the other convective events, at least for the XPORT radar. For the stratiform event, the ML extends between 800 and 1500 m asl during the first part of the event (between January 3 20:00 UTC to January 4 01:30 UTC) and then rises in about 2 hours to stabilize at an altitude range of about 2200-2800 m asl after 04:00 UTC, passing progressively at the level of the
MOUC radar in the meantime.

*Figure 2 here*

As an additional illustration of the dataset, Fig. 3 gives two examples of XPORT PPIs at 7.5° elevation angle for moderate (left) and intense (right) rain during the July 21, 2017 event. As a clear feature, one can see that, for this elevation angle, the
radar beam is fully blocked by the Chartreuse mountain range in the northern sector. Also visible in the north-east sector and, to a lesser extent, in the south-west sector are partial beam blockages associated with tall trees in the vicinity of the XPORT radar on the Grenoble campus. This figure is also intended at drawing the attention of the reader on the decrease on the Chamrousse and Moucherotte mountain returns (within red circles) during the intense rain time step compared to their values in moderate rain, as a first illustration of the MRT principle.


*Figure 3 here*

### 3. Methodology

Our aim is to study the relationship between two radar observables of propagation effects at X-Band: path-integrated
attenuation and differential propagation phase due to precipitation occurring along the radar path. We describe in the following
two sub-sections the estimation methods that were implemented. In sub-section 3.3, we complement the methodology
description by the presentation of DSD-derived $k - K_{dp}$ relationships.

### 3.1. Path-integrated attenuation estimation

Let us express the PIA (in dB) at a given range $r$ (km) as:

$$PIA(r) = PIA(r_0) + 2 \int_{r_0}^{r} k(s) \, ds \tag{1}$$

where $k(s)$ (dB km⁻¹) is the specific attenuation due to rain at range $s$ (km). $r_0$ is the range where the measurements start to
become exploitable, i.e. the range where measurements are free of ground clutter associated with side lobe effects. The term
$PIA(r_0)$ represents the so-called on-site attenuation resulting from radome attenuation and range attenuation at range closer
than $r_0$. Note that PIAs can be obtained from eq.1 for both the horizontal and the vertical polarizations. In the present article,
we will restrict ourselves to the horizontal polarization, the study of differential attenuation being a possible topic for a future
study. Delrieu et al. (1999) have proposed an assessment of the quality of PIA estimates from mountain returns by
implementing a receiving antenna in the Belledonne mountain range in conjunction with an X-band radar operated on the
Grenoble campus. They found a good agreement between the two PIA estimates for PIAs exceeding the natural variability of
the mountain reference target during dry weather. They recommended using strong mountain returns (greater than e.g. 50 dBZ
during dry weather) so as to minimize the impact of precipitation falling over the reference target itself. They also point out
that this approach is not able to separate the effects of on-site and range attenuation. They verified however, by implementing
the receiving antenna close to the radar (at a range of about 200 m), that the on-site attenuation was negligible for a radomeless
radar, which is the case for the XPORT radar but not for the MOUC radar. Another interesting feature of the MRT PIA
estimator is its independence with respect to eventual radar calibration errors.

In the current study, we used the following procedure to determine the mountain reference targets for the XPORT radar:

A large series of raw reflectivity data, observed during widespread rainfall with no ML contamination, was accumulated and
averaged in order to characterize the detection domain of the XPORT radar at the 7.5° elevation angle. This allowed us to
determine the mountain returns, the full beam blockages due to mountains, the partial beam blockages due to tall trees as well
as spurious detections due to side-lobes in the vicinity of the radar. A manual selection of the mountain reference targets was
then performed based on the map of the apparent reflectivity above 45 dBZ. The targets, made of mountain returns from

successive radials (up to 9) with a limited range extent (less than 2.0 km), are described in Table 3. Based on the radar equation and the receiver characteristics, care was taken to discard targets eventually subject to saturation at close range. The selected targets are located at a mean range comprised between 4.1 and 17.1 km, and have sizes between 0.06 and 0.94 km². For each rain event, dry-weather data before and/or after the event were used to characterize the mean target reflectivity and its time variability. Note that the mean reflectivity for each target and each time step was computed as the average of the dBZ values of each radial gate composing the target. This is justified by the fact we aim at estimating PIAs in dB. Table 3 lists the mean, standard deviation, 10 and 90% quantiles of the time series of the dry-weather apparent reflectivity of the reference targets for the first and last event of the considered series. One can notice the good stability of the mean reflectivity values between the two events, an indication of both the radar calibration stability during the period and a moderate impact of the mountain surface conditions, already evidenced in previous studies (e.g. Delrieu et al. 1999; Serrar et al. 2000) in similar mountainous contexts. The standard deviations of the reflectivity time series range between 0.2 to 0.9 dBZ, and the mean 10-90% inter-quantile range is equal to 1.03 dBZ.

Due to limited data availability, a simpler approach was implemented for the selection of the MOUC mountain reference targets. Here again, the raw reflectivity data were accumulated and averaged, but only over the period January 3$^{rd}$, 2018, 19:00 – 23:55 UTC preceding the rise of the ML at the level of the MOUC radar. It was snowing during this period at the MOUC radar site. So, we are implicitly making the assumption of negligible attenuation during snowfall (supported in the literature, e.g. Matrosov et al. 2009) in the considered case study. Table 4 displays the geometrical characteristics of the targets, as well as the mean, standard deviation, 10% and 90% quantiles of their apparent reflectivity time series. Targets are located at greater distances than those of the XPORT radar, i.e. between 19.9 and 44.9 km. In spite of having larger sizes (between 0.7 and 4.0 km²), this range effect probably explains their standard deviations to be higher, between 0.75 and 1.44 dBZ. The 10-90% inter-quantile ranges are subsequently higher as well, with a mean value of 2.6 dBZ.

The top graphs of Fig. 4 give two examples of apparent reflectivity profiles for a radial of a given target, during the July 21$^{st}$, 2017 rain event. The example on the left side corresponds to a moderate PIA (5.4 dB when considering all the gates of the radials composing the target) and the right-side example corresponds to one of the highest PIA value observed (27.6 dB) in our dataset. We tried to limit as far as possible the radial extent of targets (less than 2000 m) and/or multi-peaks targets, such as the one shown on the left-side example, in order to limit positive bias on MRT PIA estimates. The top graphs of Fig. 5 give two examples of apparent reflectivity time series during the events of July 21$^{st}$, 2017 and July 20$^{th}$, 2018, together with the mean, 10% and 90% quantiles of the dry-weather apparent reflectivity. For both cases, the XPORT data acquisition started a bit after the actual beginning of the storm. Therefore, the dry-weather reference values were estimated with data collected after the event, between 19:00 and 22:00 UTC for the July 21$^{st}$, 2017 event and between 00:00 and 06:00 UTC the day after for the July 20$^{th}$, 2018 event. For these convective events, one can note the erratic nature of the apparent reflectivity time series at the XPORT radar acquisition period used at that time (about 7 min). The MRT PIA estimates are simply calculated as the

difference between the mean values of the target apparent reflectivity during dry-weather and at each time step of the rain event (blue lines in the bottom graphs of Fig. 5).

### 3.2. Differential propagation phase estimation

Let us express the total differential phase shift between co-polar (hh and vv) received signals as:

$$\psi_{dp}(r) = 2 \int_{r_0}^{r} K_{dp}(s) \, ds + \delta_{hv}(r) \qquad (2)$$

where $K_{dp}(s)$ is the specific differential phase shift on propagation [° km$^{-1}$] related to precipitation at any range $s$ between $r_0$ and $r$, and $\delta_{hv}(r)$ is the differential phase shift on backscatter [°] at range $r$.

The quantity of interest, the differential propagation phase associated with precipitation along the path, is denoted:

$$\phi_{dp}(r) = 2 \int_{r_0}^{r} K_{dp}(s) \, ds = \psi_{dp}(r) - \delta_{hv}(r) \qquad (3)$$

As with the on-site attenuation for the MRT technique, we have here a problem with the possible influence of the differential
phase shift on backscatter $\delta_{hv}(r)$ that may introduce a positive bias on the estimation of the differential phase shift associated with precipitation along the path. We find in the literature (e.g. Otto and Russenberg 2011; Schneebeli and Berne 2012) power-law relationships between $\delta_{hv}$ and $Z_{dr}$ at X-band in rain, giving values for differential phase shift on backscatter in the ranges of [0.6° – 1.0°] and [2.1° – 3.5°] for differential reflectivity of 1 and 2 dB, respectively. Scattering simulations based on disdrometer data (Trömel et al. 2013) indicate that there may exist quite a large scatter with respect to such power-law models
and an important influence of the considered hydrometeor temperature. From simulations based on radar data at various frequencies, the same authors quantify $\delta_{hv}(r)$ values as high as 4° in the ML at X-Band and mention that strong $\delta_{hv}(r)$ values may be associated with both large dry hailstones and wet hailstones especially at X-Band. Let us note that no hail was reported for the convective cases considered in the present study. Keeping the related orders of magnitude in mind and the fact that significant $\delta_{hv}$ effects are associated with "bumps" in the $\psi_{dp}$ profiles, we will carefully discuss hereafter the possibility to
assume $\delta_{hv}$ to be negligible or not with respect to $\phi_{dp}$.

In this study, the following method was implemented for the processing of the $\psi_{dp}$ profiles and the subsequent estimation of $\phi_{dp}$ values near the mountain targets for the XPORT radar (rain case based on convective events):

We first determined so-called "rainy range gates" along the path by using the $\rho_{hv}$ profiles. The raw $\psi_{dp}(r)$ values for which $\rho_{hv}(r)$ was less than 0.95 (empirical threshold with limited impact in the [0.95-0.97] range) were set to missing values. In
addition, we defined the beginning of the rainy range by determining the first series of 10 successive gates (again an empirical

choice, corresponding to a range extent of 342 m) overpassing this threshold. The $r_0$ value was set to the minimum range value of this series. Similarly, we defined the end of the rainy range by determining the last series of 10 successive range gates overpassing this threshold close to the mountain target. A maximum rainy range, denoted $r_M$, was defined as the maximum range value of this series. It is noteworthy to mention that rain likely occurs in the ranges less than $r_0$ and greater than $r_M$, as well as in the intermediate ranges for which the $\psi_{dp}(r)$ values were set to missing values. It is however critical to discard such gates that may be prone to clutter due to side lobes close to the radar or to mountain returns close to the mountain target. Although the intermediate missing values will not impact the $\phi_{dp}$ estimation, we have to mention that both the initial and final missing values may result in a negative bias on the PIA estimation based on $\phi_{dp}(r_M)$.

In the current version of the procedure, every single radial was processed separately. First, an unfolding was applied by adding 360° to negative $\psi_{dp}(r)$ values. The system differential phase shift was estimated as the median of the $\psi_{dp}(r)$ values corresponding to the beginning of the rainy range. This value was substracted to the raw $\psi_{dp}(r)$ profiles, and eventual negative values were set to 0. Regarding the $\psi_{dp}$ measurement noise processing, we have implemented and improved a regularization procedure initially proposed by Yu and Gaussiat (2018). This procedure consists in defining an upper envelope curve, starting from $r_0$, and a lower envelope curve, starting from $r_M$, by considering a maximum jump, denoted *diffmax*, authorised between two successive gates. The calculation was performed for a series of *diffmax* values in the range of 0.5 – 10°. The regularized $\psi_{dp}$ profiles (increasing monotonous curves) were estimated by taking the average of the upper and lower envelope curves. Note that the values for the missing gates between $r_0$ and $r_M$ were simply interpolated with the adjacent values of the regularized profile. A mean absolute difference criterion (MAD) between the raw and regularized profiles over a series of 30 gates with non-missing values near the mountain target (empirical choice, corresponding to a range extent of about 1 km) was used to determine the optimal *diffmax* value and the associated profile. The optimal profile was finally selected if the MAD criterion was less than 50%, otherwise we considered the polarimetry-derived PIA to be missing for the considered radial. Finally, the $\phi_{dp}(r_M)$ value for the target was estimated as a weighted average of the $\phi_{dp}(r_M)$ values of all the non-missing radials composing the target, the weights being the number of reference gates of each radial. The bottom graphs of Fig. 4 present the raw and regularized profiles, as well as the envelope curves, for the examples already commented above. For the right-hand example corresponding to one of the strongest PIA (27.6 dB) observed, one can note that the noise of the raw $\psi_{dp}$ profile is low, especially in the range with the highest gradients between 7 and 13 km. There is no apparent "bump" on the raw profile that could sign a $\delta_{hv}$ contamination; so that one might be tempted to consider the regularized profile as a good estimator of the $\phi_{dp}$ profile in that case. The left-hand side example, corresponding to a moderate MRT-derived PIA of 5.4 dB, is more complex. As already noted, the mountain target itself is noisy with significant mountain return contamination before range $r_M$ as evidenced by the $\rho_{hv}$ profile. In addition, one can note a non-monotonic behaviour of the raw $\psi_{dp}$ profile with a plateau of about 17.5° for ranges greater than 4 km, following an increase in the raw profile (with moderate noise) up to 22° at 4-km range. One might assume a $\delta_{hv}$ contamination in that case. Interestingly, the regularisation procedure is shown

to provide a good filtering of the "bump", and here again we are tempted to consider the regularized profile as a good estimator of the $\phi_{dp}$ profile. The middle graphs in Fig. 5 display the time series of the $\phi_{dp}(r_M)$ values associated with the apparent reflectivity of mountain returns discussed above. One can note a good consistency of the two time series for the highest peaks while discrepancies can be evidenced for the moderate and small values.

Basically, the same methodology was implemented for the MOUC radar case study, with some alterations to be described hereafter. Figure 6 provides the time series of the apparent reflectivity of a given mountain target, the resulting PIA estimates and the $\phi_{dp}(r_M)$ estimates for the 0°-PPI of the MOUC radar during the stratiform event of January 3-4th, 2018. The time period considered in the figure ranges from 00:00 UTC to 06:00 UTC on January 4th, 2018 in order to focus on the rising of the ML between 02:00 UTC and 04:00 UTC. The target is located at a distance of 19.9 km from the radar. The bottom graph of Fig. 6 displays the results of the ML detection algorithm (Khanal et al. 2019) in terms of the altitudes of the top, peak and bottom of the $Z_h$ (blue) and the $\rho_{hv}$ (orange) ML signatures. The altitude of the $Z_h$ top inflexion point is assumed to correspond to the 0°C isotherm altitude while the $\rho_{hv}$ bottom inflexion point corresponds well with the bottom of the ML according to Khanal et al. (2019). We therefore define the ML width as the altitude difference between Zh top and $\rho_{hv}$ bottom. Before 02:00 UTC, the ML is well below the altitude of the MOUC radar. MOUC radar measurements at the 0°-elevation angle are therefore made in snow/ice precipitation during this period. Based on the ML detection results, the passage of the ML at the altitude of the MOUC radar begins at about 02:20 UTC and ends at 04:10 UTC. After this time, MOUC radar measurements are therefore made in rainfall.

*Figure 6 here*

As representative examples, Fig. 7 illustrates range profiles taken by the MOUC radar during the snowfall (left) and the ML (right) periods. As expected, the $\rho_{hv}$ profiles are very different in the two cases, with $\rho_{hv}$ values close to 1 in snow indicating precipitation homogeneity while $\rho_{hv}$ presents a high variability in the ML. During the ML period, we had therefore to adapt the $\rho_{hv}$ threshold used to detect gates with precipitation. Based on the $\rho_{hv}$ peak statistics presented by Khanal et al. (2019), we have chosen a value of 0.8. As it can be seen in Fig. 7, such a threshold may prevent detection of the mountain reference return itself. Subsequently, we had to adapt the determination of ranges $r_0$ and $r_M$ with respect to the XPORT radar case, firstly by considering two successive gates corresponding to a range extent of 480 m (instead of 10 gates, corresponding to 342 m) and secondly by making sure that the calculated $r_M$ value was less than the range of the first mountain reference gate. Regarding the regularization of the $\psi_{dp}$ profiles (bottom graphs of Fig. 7), it was found that the raw profiles were noisier compared with the XPORT case study. Well-structured "bumps" were not evidenced in the ML profiles, maybe as a result of the lower range resolution of the MOUC radar, and the regularisation procedure was found to work satisfactorily. It remains however difficult to assume that there is no $\delta_{hv}$ contamination during the ML period.

*Figure 7 here*

Coming back to Fig. 6, one can note the mean value of $\phi_{dp}(r_M)$ to be equal to 11.2° during the snowfall period, resulting in a specific differential phase shift on propagation of 0.28 °km$^{-1}$ if the differential phase shift on backscatter is neglected. Such values indicate a significant heterogeneity of the horizontal and vertical dimensions of the snow/ice hydrometeors. During the rainy period between 04:10 and 06:00 UTC, there is a good coherence between the specific attenuations derived from the MRT PIA (0.078 dB km$^{-1}$ at around 04:00 UTC - 0.035 dB km$^{-1}$ at 06:00 UTC) and those derived from the polarimetry (0.076 - 0.046 dB km$^{-1}$ at the same time steps) using the $k - K_{dp}$ relationship established for this event by using the DSD measurements available at the IGE site (see section 3.3 below).

Our main objective with the January 3-4$^{th}$, 2018 event is to study the $\phi_{dp} - PIA$ relationship within the ML. Figure 6 indicates that both variables take, as expected, higher values during that period compared to during the snowfall and the rainfall periods. The maximum values reached are 14.2 dB for PIA and 25.6° for $\phi_{dp}(r_M)$. Figures 6b and 6c also show that the co-fluctuation of the two time series is not that good during the ML period with a $\phi_{dp}(r_M)$ signal having a trapezoidal shape with maximum values between 02:35 UTC and 03:15 UTC while the MRT PIA signal is more triangular and peaks at 03:15 UTC. We note that the two signals compare well after the peak and that they both peak down at 03:55 UTC when measurements are made in the lowest part of the ML. These features are quite systematic for all the thirteen targets considered for the MOUC radar for this event, giving the impression that the $\phi_{dp} - PIA$ relationship depends on the position within the ML and as such on the physical processes occurring during the melting. This will be further illustrated and discussed in sub-section 4.2. However, we have to mention here three points that may limit the validity of such inferences for the MOUC radar configuration compared to the XPORT one: (i) the MRT PIA estimates may be positively biased by radome attenuation, (ii) the polarimetry – derived PIA estimates may be affected by $\delta_{hv}$ contamination in the ML and (iii) non-uniform beam filling effects become probably significant for the 20-40 km range considered, leading to a smoothing of the radar signatures. There is no evidence so far of the first two points in the available dataset; this may be due to the moderate intensity of this precipitation event.

### 3.3. Study of the $k - K_{dp}$ relationship in rain from in-situ DSD measurements

Before presenting the analysis of the $\phi_{dp} - PIA$ relationship in rain and in the melting layer based on the estimates for all the mountain targets and time steps available for the two sets of events, we study in this sub-section the $k - K_{dp}$ relationships that we were able to derive from the DSD measurements collected at ground level at the IGE site. For all the events, precipitation was in the form of rainfall at this altitude. As for the scattering model, we used the CANTMAT version 1.2 software programme that was developed at Colorado State University by C. Tang and V.N. Bringi. The raw PARSIVEL2 DSD measurements have a time resolution of 1 min. The volumetric concentrations were computed with a 5-min resolution and binned into 32 diameter classes with increasing sizes from 0.125 mm up to 6 mm. The CANTMAT software uses the T-Matrix formulation to compute radar observables such as horizontal reflectivity, vertical reflectivity, differential reflectivity, co-polar cross-correlation, specific attenuation, specific phase shift, etc, as a function of the DSD, the radar frequency, air

temperature, oblateness models (e.g. Beard and Chuang 1987; Andsager et al. 1999; Thurai and Bringi 2005) and canting models for the rain drops as well as the incidence angle of the electromagnetic waves. Figure 8 displays the empirical $k - K_{dp}$ pairs of points obtained for the convective events (left) and the stratiform one (right) as well as the fits of least-square linear models and power-law non-linear regressions.

*Figure 8 here*

Based on the literature review mentioning an almost linear relationship between $k$ and $K_{dp}$ at X-Band (Bringi and Chandrasekar, 2001; Testud et al. 2000; Schneebeli and Berne 2012) we have first tested a linear regression with an intercept

forced to be equal to 0 (red lines in Fig. 8). This simple model indeed provides a rather good fit to the data, especially for the convective events. Due to the observed bending of the scatterplots, we have also tested a non-linear regression to a power-law model (blue curve) which significantly improves the fittings. A sensitivity analysis was performed in order to test the influence of the raindrop temperature, the raindrop oblateness model, the standard deviation of the canting angle distribution, the incidence angle. For reasonable ranges of variation of these parameters, the DSD itself appears to be the most influent factor

on the values of the regression coefficients. We note that the slopes of our 0-forced linear models are significantly higher than values proposed in the literature (0.233 in Bringi and Chandrasekar (2001); 0.205 – 0.245 in Scheebeli and Berne (2012)). The exponents of the fitted power-law models are also significantly higher than 1.0. The fits in Fig. 8 correspond to the most likely parameterization of the scattering model in terms of temperature and incidence angles for the two events, i.e. 20°C and 7.5° respectively for the convective cases and 0°C and 0° for the stratiform case. The Beard and Chuang (1987) formulation was

used as the raindrop oblateness model. The DSD-derived linear and non-linear $k$ - $K_{dp}$ relationships were used to process the regularized $\phi_{dp}(r)$ profiles which were first simply derivated to obtain the $K_{dp}(r)$ profiles prior to the application of the two $k$ - $K_{dp}$ relationships. The bottom graphs of Fig. 5 shows examples of the resulting polarimetry-derived PIAs.

**4. Results**

**4.1. Study of the $\phi_{dp} - PIA$ relationship in rain**

Figure 9 displays the scatterplot of the $\phi_{dp} - PIA$ values obtained for the nine convective events (Table2) with the XPORT 7.5°-PPI data, following the methodology described in sections 3.1 and 3.2. The data from the sixteen mountain targets (Table

3) were considered. For a given event, targets with maximum MRT-derived PIAs less than 5 dB were discarded in order to limit the weight of small PIA estimates in the global analysis. Since we consider the two variables on an equal footing, we preferred to calculate the least-rectangles regression (blue straight line) between the two variables rather than the least-squares regression of one variable over the other one. One can notice the rather large dispersion of the scatterplot, with explained

variance of 77%. We note the regression slope (0.41) to be higher than the slope of the $k - K_{dp}$ linear relationship (0.336),
reported as the red straight line in Fig. 9.

*Figure 9 here*

To go further, Fig. 10 presents the comparison of the MRT-derived PIAs with the polarimetry-derived PIAs. The linear $k - K_{dp}$ relationship leads to a significant positive bias for the polarimetry-derived PIAs with a least-rectangles slope of 1.24. The non-linear $k - K_{dp}$ relationship does indeed a good job in reducing this bias (least-rectangles slope of 1.03). This result may be surprising given the $k - K_{dp}$ relationships displayed in Fig. 8. One has to realize that the range of $K_{dp}$ values is much smaller for the 5-min DSD estimations than for the $K_{dp}(r)$ profiles discretized with a 34.2 m resolution. Considering the 1-min DSDs allowed us to confirm the validity of the linear and non-linear $k - K_{dp}$ models for a wider $K_{dp}$ range (not shown here for the sake of conciseness). We are therefore confident in the relevance of the results presented in Fig. 10.

*Figure 10 here*

**4.2. Study of the $\psi_{dp} - $ PIA relationship in the Melting Layer**

Figure 11 displays the scatterplot of the $\phi_{dp} - PIA$ values obtained in the ML for the January 4[th], 2018 stratiform event with the MOUC 0°-PPI data, following the methodology described in section 3.1 and 3.2. The results obtained for the thirteen targets (Table 4) are considered in this analysis, with no target censoring based for instance on the minimum PA observed for a given target as for the XPORT case study. One can see that the correlation between the two variables is severely degraded compared to the rain case with an explained variance of 41% and a least-rectangle slope of 0.51 dB degree[-1]. The red line recalls the $k - K_{dp}$ linear regression determined with the DSD observed at ground level for this event. Clearly, the $\phi_{dp} - PIA$ relationship is different in rain and in the ML, and as suggested when commenting Fig. 6, it likely depends on the physical processes occurring during the melting.

*Figure 11 here*

To investigate this point, $\phi_{dp}(r_M)$ and $PIA(r_M)$ values estimated during the rising of the ML at the level of the MOUC radar are represented in Fig. 12 as a function of their position within the ML. As already noted, we define the ML width as the difference between the Zh top altitude and the $\rho_{hv}$ bottom altitude (Khanal et al. 2019). Since the ML width significantly varies during the considered period (from 630 to 1020 m; see Fig. 8), we found necessary to scale the altitudes by the ML width. This was achieved by considering the following linear transformation of the altitudes:

$$H(t) = (h_M - h_{\rho hvB}(t))/MLw(t) \qquad (4)$$

where $h_M$ is the altitude [m asl] of the MOUC radar, $h_{\rho hvB}(t)$ is the altitude of the ML bottom and $MLw(t)$ is the ML thickness at a given time $t$. The scaled altitude $H(t)$ [-] subsequently takes the value 0 at ML bottom and the value 1 at ML top (orange and blue thick horizontal lines, respectively, in Fig. 12). Furthermore, in order to locate more precisely the position of the Zh and $\rho_{hv}$ peaks within the ML, we computed their scaled altitudes at each time step, $H_{zhP}(t)$ and $H_{\rho hvP}(t)$ respectively, as:

$$H_{zhP}(t) = (h_{zhP}(t) - h_{\rho hvB}(t))/MLw(t) \qquad (5)$$

and:

$$H_{\rho hvP}(t) = (h_{\rho hvP}(t) - h_{\rho hvB}(t))/MLw(t) \qquad (6)$$

where $h_{zhP}(t)$ and $h_{\rho hvP}(t)$ are the altitudes of Zh peak and $\rho_{hv}$ peak at time $t$. The dotted horizontal lines in Fig. 12 represent the 10 and 90% quantiles of the time series of the scaled altitudes of Zh peak (dotted blue lines) and $\rho_{hv}$ peak (dotted orange lines). We can observe a shift between the Zh and $\rho_{hv}$ characteristic altitudes, consistent with the ML climatology established by Khanal et al. (2019) who reported a shift of about 100 m in average between the two peaks. We note in Fig. 8 that this shift is visible during the snowfall period and at the beginning of the ML rising but that it is less pronounced after 03:00 UTC and during the rainfall period. In order to better evidence their vertical trends, the MRT $PIA(r_M)$ and $\phi_{dp}(r_M)$ values are presented in Fig. 12 as a function of the scaled altitudes in the form of box plots with a scaled altitude class of size 0.1. The number of counts in each class is indicated on the right of the graphs; it is a multiple of the number of MRT targets (13 here) depending on the time occurrence of estimates in a given altitude class. The vertical sampling is not very rich, with missing classes within the ML. However there is clear signature for the two variables in the ML. The trends already evoked when commenting Fig.8 are confirmed: (i) the MRT PIAs peak when measurements are made at the level of the Zh and $\rho_{hv}$ peaks; more precisely, the PIA peak is observed for the altitude class containing the $\rho_{hv}$ peaks (scaled altitude class centered at 0.3); (ii) the region with maximum values is somewhat thicker for $\phi_{dp}$, encompassing a significant part of the upper ML, between the 0.3 and 0.8 scaled altitude classes; (iii) $\phi_{dp}$ tends towards almost similar values in average in rain (ML bottom) and snow (ML top), (iv) the PIA tends towards its value in rain below the ML and towards 0 above the ML. One would have expected a more pronounced return towards 0 of the PIA on top of the ML. This lower than expected decrease could sign a radome attenuation; however the rainfall intensity is low for the considered event and the radome is equipped with a heating system so that accumulated snow is unlikely. It may also result from a smoothing effect related to non-uniform beam filling: with its 3-dB beamwidth of 1.28°,

the angular resolution of the measurements of the MOUC radar is 447 m and 1005 m at distances of 20 km and 45 km,
respectively, which correspond to the minimum and maximum ranges of the considered mountain targets.

*Figure 12 here*

Finally, Figure 13 displays the evolution of the ratio of the mean of the MRT $PIA(r_M)$ values over the mean of $\phi_{dp}(r_M)$ values
as a function of the scaled altitudes. The value of the ratio below the ML (0.33) is in rather good agreement with the slope of
the linear model established between the specific attenuation $k$ and the specific differential phase shift $K_{dp}$ using the DSD
measurements in rain available for this event (0.29, see Fig. 9). Near the $\rho_{hv}$ peak, the ratio value is equal to 0.42. For the three
classes of scaled altitude 0.7, 0.8 and 0.9, the ratio is between 0.32 and 0.38, with an apparent secondary maximum for the
altitude class 0.8. Data with increased vertical resolution would be necessary to confirm or not this observation, which is also
visible on the PIA profile and on several $\phi_{dp}$ and $PIA$ time series like the ones displayed in Fig. 8. Above the ML, the ratio
progressively tends toward 0 in about 300 to 400 m.

*Figure 13 here*

**5. Summary and conclusions**

We developed in this work a methodology for studying the relationship between total differential phase shift ($\phi_{dp}$) and path-
integrated attenuation ($PIA$) at X-Band. Knowledge of this relationship is critical for the implementation of attenuation
corrections based on polarimetry. We used the Mountain Reference Technique for direct PIA estimations associated with the
465 decrease of strong mountain returns during precipitation events. The MRT sensitivity depends on the time variability of the
dry-weather mountain returns. The MRT PIAs may be positively biased by on-site attenuation related in particular to radome
attenuation and negatively biased by the effect of precipitation falling over the reference targets. The polarimetry PIA
estimation is based on the regularization of the raw $\psi_{dp}$ profiles and their derivation in terms of specific differential phase
shift ($K_{dp}$) profiles followed by the application of a power-law relationship between the specific attenuation and the specific
differential phase shift. Such $k - K_{dp}$ relationships were evaluated for rain with a scattering model by using DSD
measurements and an oblateness model for raindrops. The noise of the raw $\psi_{dp}$ profiles, the possible contamination of the
signal by differential shift on backscatter and the adequacy of the $k - K_{dp}$ relationship used determine the quality of the
polarimetry derived PIAs. Non-uniform beam filling (NUBF) effects may also play a role. A point to emphasize is that both
PIA estimators are not sensitive to an eventual radar miscalibration.

We presented first a rain case study based on nine convective events observed with the XPORT radar located in the Grenoble valley. Sixteen mountain targets were considered with dry-weather mean apparent reflectivity greater than 45 dBZ. The stability of the apparent reflectivity of the mountain targets was shown to be very good, an indication of a good radar calibration stability during the considered period. The time variability of the reference returns during dry-weather preceding or succeeding the rain events was also found to be very small with standard deviations in the range of [0.2 – 0.9 dBZ], enabling a MRT PIA sensitivity better than 1 dB. Since the XPORT radar is radomeless, on-site attenuation effects are most likely negligible. The impact of rain falling over the mountain targets may also be very limited due to the high reflectivity threshold considered (45 dBZ). The development of the regularization procedure of the raw $\psi_{dp}$ profiles required a significant effort and we are confident in its ability at dealing with the measurement noise, especially for heavy precipitation. We carefully examined many raw and regularized profiles looking for possible evidence of $\delta_{hv}$ contamination during the considered convective events. We found out some profiles with rather well organised "bumps" that could sign such contaminations. The regularization procedure was adapted in order to filter such effects, with a satisfactory performance when they occur at some distance (some kilometres) from the mountain target. In addition, we remind the reader that the observed $\psi_{dp}(r_M)$ values extends up to 80° while the theoretical $\delta_{hv}$ range is 0-4 dB. The $\delta_{hv}$ effect may therefore impact the results obtained only at the margin in the considered case study. NUBF effects may constitute an additional source of error which, although the rain events were convective, should remain limited due to the short ranges considered. In the end, the scatterplot of the MRT PIAs as a function of the $\phi_{dp}(r_M)$ values for all the nine convective events presents an overall good coherence with however a significant dispersion (explained variance of 77%). It is interesting to note that the non-linear $k - K_{dp}$ relationship derived from independent DSD measurements taken during the events of interest at ground level allows a satisfactory transformation of the XPORT $\phi_{dp}(r_M)$ values into almost unbiased (although dispersed) PIA estimates. Both estimation methods are prone to specific errors and, even if the MRT PIA estimator is more directly related to power attenuation, it is *a priori* difficult to say which estimator is the best. An assessment exercise of attenuation correction algorithms, making use of both PIA estimators, with respect to an independent data source (e.g. raingauge measurements), is desirable to distinguish the two PIA estimators. In this perspective, a specific experiment is being designed within the RadAlp project and it will be implemented in the near future.

The Melting Layer case study of January 3-4$^{th}$ 2018 was made possible by the unique configuration of the observation system available. The study of the $k - K_{dp}$ relationship within the ML is desirable to better quantify attenuation effects in the ML with polarimetry; and one has to recognize that such relationship can still be very difficult to characterize theoretically with scattering models and particle size distributions to be collected in the ML. The XPORT radar located at the bottom of the valley allowed a detailed temporal tracking of the ML from below using quasi-vertical profiles derived from 25°-PPIs. The MOUC radar provided horizontal scans at an altitude of 1917 m asl in direction of several mountain targets during the rising of the ML in about 2 hours. From this dataset, it was possible to derive the evolution of $PIA(r_M)$ and $\phi_{dp}(r_M)$ values as a function of the altitude within the ML. The evolution with the altitude of the ratio of the mean value of $PIA(r_M)$ over the mean

value of $\phi_{dp}(r_M)$, as a proxy for the slope of a linear $k - K_{dp}$ relationship within the ML, was also considered. Since the ML

width varied during the ML rising, we found necessary to scale the altitudes with respect to the ML width. The three variables considered present a clear signature as a function of the scaled altitude. In particular, the $PIA/\phi_{dp}$ ratio peaks at the level of the $\rho_{hv}$ peak (somewhat lower than the Zh peak), with a value of 0.42 dB degree$^{-1}$, while its value in rain just below the ML is 0.33 dB degree$^{-1}$. The latter value is consistent with the slope of the linear $k - K_{dp}$ relationship (0.29) established from concomitant DSD measurements at ground level. The $PIA/\phi_{dp}$ ratio remains quite strong in the upper part of the ML, between

0.32 and 0.38 dB degree$^{-1}$, before tending towards 0 above the ML. One would have expected a more pronounced return towards 0 of the PIA on top of the ML. This lower than expected decrease could sign on-site attenuation occurring at the beginning of the ML rise due to the melting of the snow eventually accumulated over the radome; this effect is probably low for the considered event since the snowfall intensity was small and since the radome is heated. It may also result from a smoothing effect related to non-uniform beam filling (angular resolution of 447 and 1005 m for the range of mountain target

distances). The $\delta_{hv}$ effect is likely to be strong in the ML (up to 4°) and its relative importance may be quite high in our case study since the PIA range is significantly lower compared to the rain case study, with maximum PIAs of about 15 dB (note also that the sensitivity of the MRT is less than for the XPORT case study since the dry-weather variability of the mountain returns is higher with standard-deviations in the range [0.62-1.44]). However, we did not find evidence of $\delta_{hv}$ signatures in the raw $\psi_{dp}(r)$ profiles and we are confident in the ability of the regularization procedure to filter them in a rather satisfactory

way if they eventually occur. Although the experimental configuration for the study of attenuation in the ML presents some limitations (possible radome attenuation, NUBF effects), the preliminary results presented here will be deepened by processing a dataset of about thirty stratiform events with the presence of the ML at the level of the MOUC radar.

**Acknowledgements**

We are grateful to P.N. Gatlin (NASA Marshall Space Flight Center, Huntsville, AL) for providing the CANTMAT version 1.2 software developed at Colorado State University by C. Tang and V.N. Bringi, who we also thank. The RadAlp experiment is co-funded by the Labex osug@2020 of the Observatoire des Sciences de l'Univers de Grenoble, the Service Central Hydrométéorologique et d'Appui à la Prévision des Inondations (SCHAPI) and Electricité de France / Division Technique Générale (EDF/DTG).

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

**Table 1. Characteristics of the XPORT and MOUC radar systems**

| | MOUC radar | XPORT radar |
|---|---|---|
| Longitude (decimal degrees) | 5.639237 | 5.762327 |
| Latitude (decimal degrees) | 45.147736 | 45.194150 |
| Altitude (m asl) | ground: 1901<br>antenna feedhorn: 1917 | ground: 213<br>antenna feedhorn: 228 |
| Frequency (GHz) | 9.420 | 9.400 |
| Antenna diameter (m) | 1.8 | 1.8 |
| 3-dB beamwidth (°) | 1.28 | 1.37 |
| Antenna gain (dB) | 42 | 42 |
| Radome | yes | no |
| Peak power | 30 kW, on each polarisation | 50 kW, on each polarisation |
| Pulse length (µs) | 2 | 1 |
| Radial bin size (m) | 240 | 34.2 |
| Receiver dynamic range (dB) | >90 | >90 |
| Minimum detectable signal (dBm) | -114 | -112 |
| Volume scanning protocol<br>(PPIs with elevation angles in °) | 0 / 0.6 / 1.2 / 2 / 3 / 4 / 8 / 14° | 3.5 / 7.5 / 15 / 25 / 45° |
| Volume scanning period (min) | 5 | ~7 |
| Measured parameters | $Z_h, Z_v, Z_{dr}, \rho_{hv}, \phi_{dp}, v_r$ | |


**Table 2. Some characteristics of the nine convective events considered in the study of the $\phi_{dp} - PIA$ relationship in rain. The ML detection was performed with the 25°-elevation angle measurements of the XPORT radar using the algorithm described in Khanal et al. (2019). The total rain amount and the maximum rainrate are recorded at the weather station available at the XPORT radar site at IGE. The maximum PIA is derived from the MRT technique by considering the 7.5° elevation data of the XPORT radar.**


| Date | Beginning (UTC) | End (UTC) | Minimum altitude of the ML bottom (m asl) | Total rain amont (mm) | Maximum rainrate in 10 min (mm h$^{-1}$) | Maximum MRT PIA (dB) |
|---|---|---|---|---|---|---|
| May 12, 2017 | 12:00 | 16:00 | 2000 | 9.2 | 8.4 | 14.2 |
| July 21, 2017 | 15:30 | 19:30 | 3000 | 35.2 | 42.0 | 30.7 |
| August 8, 2017 | 08:30 | 14:30 | 3700 | 27.9 | 48.0 | 30.1 |
| August 31, 2017 | 07:00 | 11:30 | 3200 | 19.9 | 15.5 | 7.6 |
| May 22, 2018 | 16:00 | 23:00 | 2000 | 16.9 | 8.4 | 10.2 |
| May 27, 2018 | 14:00 | 17:00 | 2700 | 6.9 | 9.9 | 6.0 |
| May 28, 2017 | 13:00 | 23:00 | 2500 | 9.8 | 9.0 | 7.7 |
| July 20, 2018 | 17:00 | 22:00 | 2700 | 12.1 | 15.6 | 19.3 |
| August 9, 2018 | 07:30 | 15:00 | 3000 | 24.8 | 8.4 | 19.2 |



**Table 3: Geometrical characteristics and apparent reflectivity statistics for the 16 mountain targets selected for the XPORT radar at an elevation angle of 7.5°. The mean, standard deviation, 10 and 90% quantiles of the apparent reflectivity time series are given for the first and last convective events in the considered period (see Table 2).**


| Target | Mean azimuth (°) | Mean range (km) | Number of gates | Size (km²) | May 12th, 2017 | | | | August 9th, 2018 | | | |
|---|---|---|---|---|---|---|---|---|---|---|---|---|
| | | | | | Mean reflectivity (dBZ) | Standard deviation (dBZ) | 10% quantile (dBZ) | 90% quantile (dBZ) | Mean reflectivity (dBZ) | Standard deviation (dBZ) | 10% quantile (dBZ) | 90% quantile (dBZ) |
| 1 | 2.9 | 4.1 | 51 | 0.06 | 48.39 | 0.21 | 48.17 | 48.73 | 48.29 | 0.23 | 48.03 | 48.62 |
| 2 | 13.2 | 4.8 | 130 | 0.18 | 52.19 | 0.80 | 51.55 | 53.17 | 52.22 | 0.66 | 51.62 | 53.27 |
| 3 | 17.5 | 5.7 | 163 | 0.27 | 51.90 | 0.29 | 51.61 | 52.25 | 52.42 | 0.50 | 51.91 | 53.14 |
| 4 | 24.0 | 8.6 | 133 | 0.33 | 51.98 | 0.51 | 51.44 | 52.80 | 51.87 | 0.40 | 51.41 | 52.39 |
| 5 | 29.0 | 14.6 | 71 | 0.30 | 49.44 | 0.55 | 48.91 | 50.01 | 50.31 | 0.59 | 49.63 | 51.10 |
| 6 | 89.5 | 17.1 | 160 | 0.79 | 53.20 | 0.38 | 52.81 | 53.59 | 52.78 | 0.43 | 52.34 | 53.53 |
| 7 | 95.3 | 14.5 | 95 | 0.40 | 54.12 | 0.23 | 53.91 | 54.30 | 53.96 | 0.21 | 53.72 | 54.20 |
| 8 | 98.4 | 13.2 | 120 | 0.45 | 51.02 | 0.50 | 50.59 | 51.67 | 52.13 | 0.39 | 51.69 | 52.67 |
| 9 | 101.2 | 13.1 | 156 | 0.58 | 48.95 | 0.23 | 48.71 | 49.18 | 49.50 | 0.12 | 49.37 | 49.66 |
| 10 | 119.7 | 12.1 | 92 | 0.32 | 49.36 | 0.21 | 49.11 | 49.59 | 50.23 | 0.12 | 50.07 | 50.39 |
| 11 | 124.8 | 11.8 | 242 | 0.82 | 51.04 | 0.53 | 50.50 | 51.84 | 52.02 | 0.32 | 51.63 | 52.43 |
| 12 | 130.1 | 11.9 | 240 | 0.82 | 51.43 | 0.90 | 50.48 | 52.63 | 54.63 | 0.52 | 54.08 | 55.30 |
| 13 | 135.1 | 12.0 | 271 | 0.94 | 50.20 | 0.87 | 49.33 | 51.48 | 53.24 | 0.78 | 52.34 | 54.35 |
| 14 | 238.8 | 11.4 | 221 | 0.73 | 52.97 | 0.67 | 52.20 | 53.59 | 52.86 | 0.58 | 52.11 | 53.60 |
| 15 | 243.8 | 10.7 | 187 | 0.58 | 52.63 | 0.59 | 51.97 | 53.46 | 53.79 | 0.35 | 53.37 | 54.23 |
| 16 | 248.8 | 10.5 | 162 | 0.49 | 53.62 | 0.41 | 53.11 | 54.02 | 52.96 | 0.37 | 52.53 | 53.50 |


**Table 4: Geometrical characteristics and apparent reflectivity statistics for the 13 mountain targets selected for the MOUC radar at an elevation angle of 0°. The mean, standard deviation, 10 and 90% quantiles of the apparent reflectivity time series are computed over the period January 3rd, 19:00 – 23:55 UTC preceding the rising of the ML at the level of the MOUC radar.**


| Target | Mean azimuth (°) | Mean range (km) | Number of gates | Size (km²) | January 3rd-4th, 2018 | | | |
|---|---|---|---|---|---|---|---|---|
| | | | | | Mean reflectivity (dBZ) | Standard deviation (dBZ) | 10% quantile (dBZ) | 90% quantile (dBZ) |
| 1 | 40.0 | 29.52 | 25 | 1.55 | 49.97 | 1.1 | 48.70 | 51.26 |
| 2 | 43.7 | 26.28 | 13 | 0.72 | 49.90 | 1.28 | 48.18 | 51.36 |
| 3 | 78.0 | 27.12 | 24 | 1.36 | 48.18 | 1.44 | 46.77 | 50.12 |
| 4 | 84.2 | 23.64 | 28 | 1.39 | 49.56 | 0.95 | 48.37 | 50.90 |
| 5 | 89.5 | 23.04 | 82 | 3.96 | 49.09 | 0.62 | 48.39 | 49.80 |
| 6 | 96.0 | 21.36 | 78 | 3.49 | 49.37 | 0.75 | 48.32 | 50.34 |
| 7 | 101.7 | 19.92 | 52 | 2.17 | 49.31 | 1.01 | 47.83 | 50.37 |
| 8 | 107.2 | 22.44 | 33 | 1.55 | 51.94 | 1.11 | 50.52 | 53.22 |
| 9 | 117.0 | 25.32 | 38 | 2.02 | 51.50 | 1.03 | 50.17 | 52.74 |
| 10 | 121.2 | 23.52 | 41 | 2.02 | 48.65 | 1.18 | 47.28 | 50.18 |
| 11 | 128.5 | 28.44 | 43 | 2.56 | 49.38 | 0.98 | 48.21 | 50.59 |
| 12 | 132.5 | 27.00 | 25 | 1.41 | 50.33 | 1.24 | 48.71 | 51.81 |
| 13 | 160.2 | 44.88 | 37 | 3.48 | 49.91 | 1.00 | 48.69 | 51.11 |



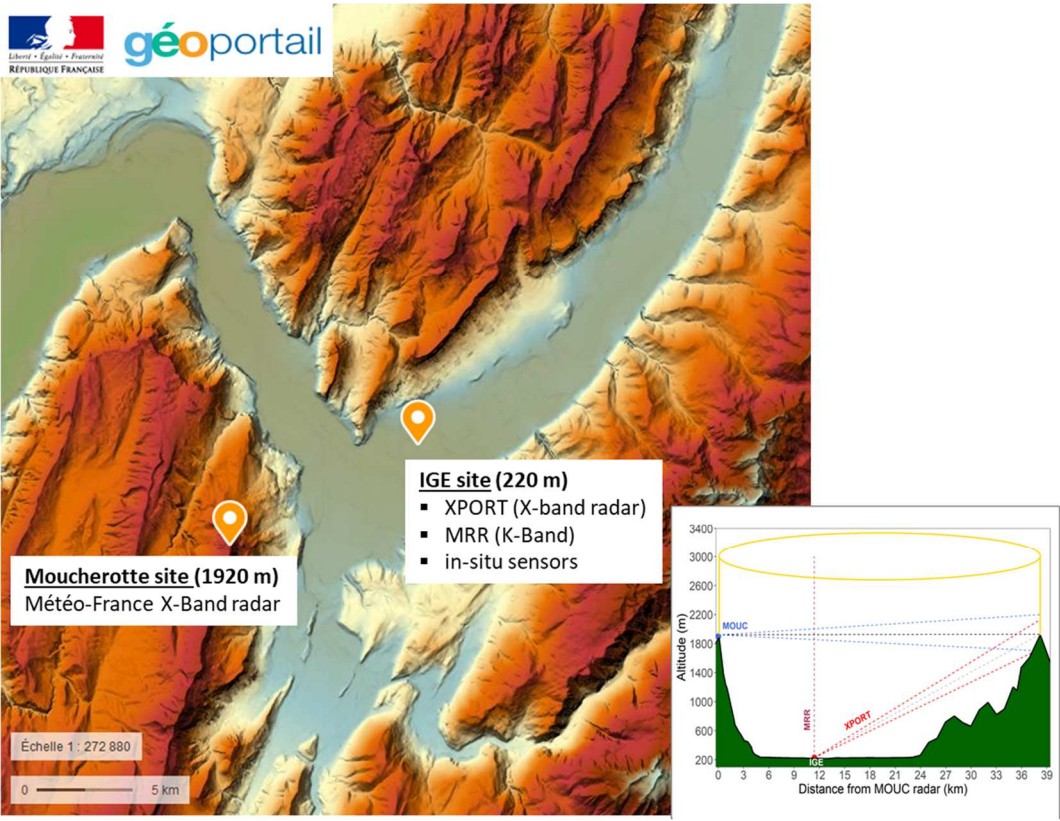


**Figure 1. The topographical map of Grenoble is shown along with positions of two radar systems. A vertical cross-section along the line joining the two radar sites is shown in the insert on the bottom right of the figure.**

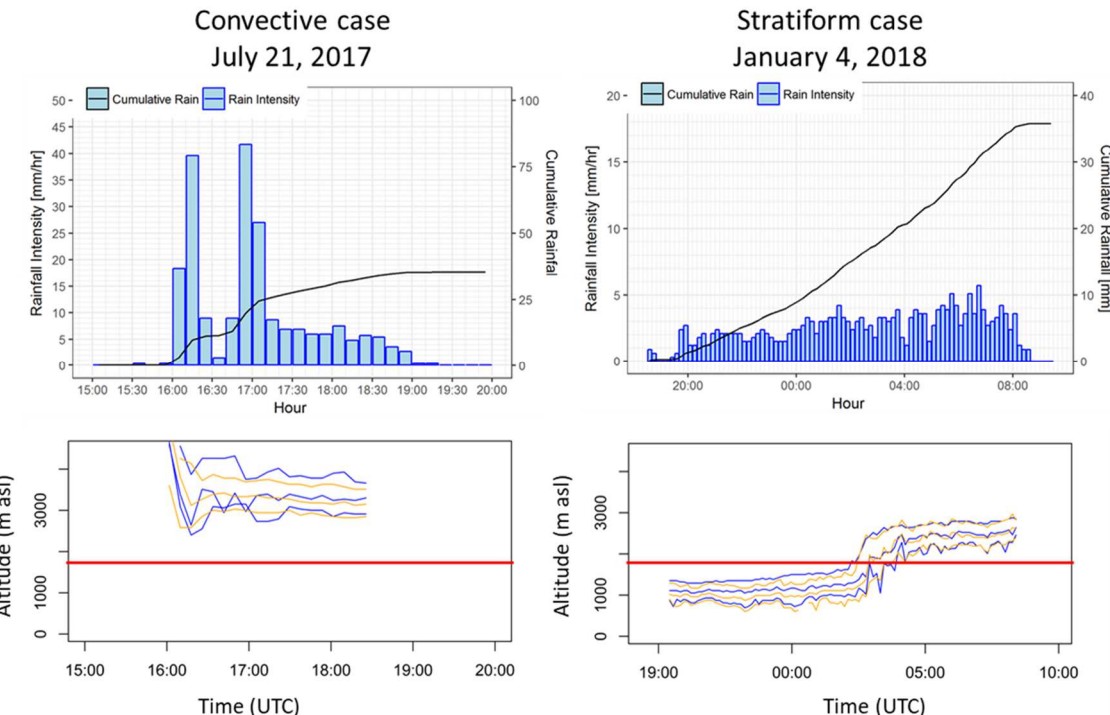

 **Figure 2. Description of two rain events considered in the present study: left – convective case of July 21, 2017; right – stratiform case of January 4, 2018. Top graphs: rainrate and cumulative rainfall timeseries observed at the IGE site; bottom: results of the ML detection algorithm based on XPORT 25°-PPI data. The horizontal red line indicates the altitude of the MOUC radar; see text for details.**

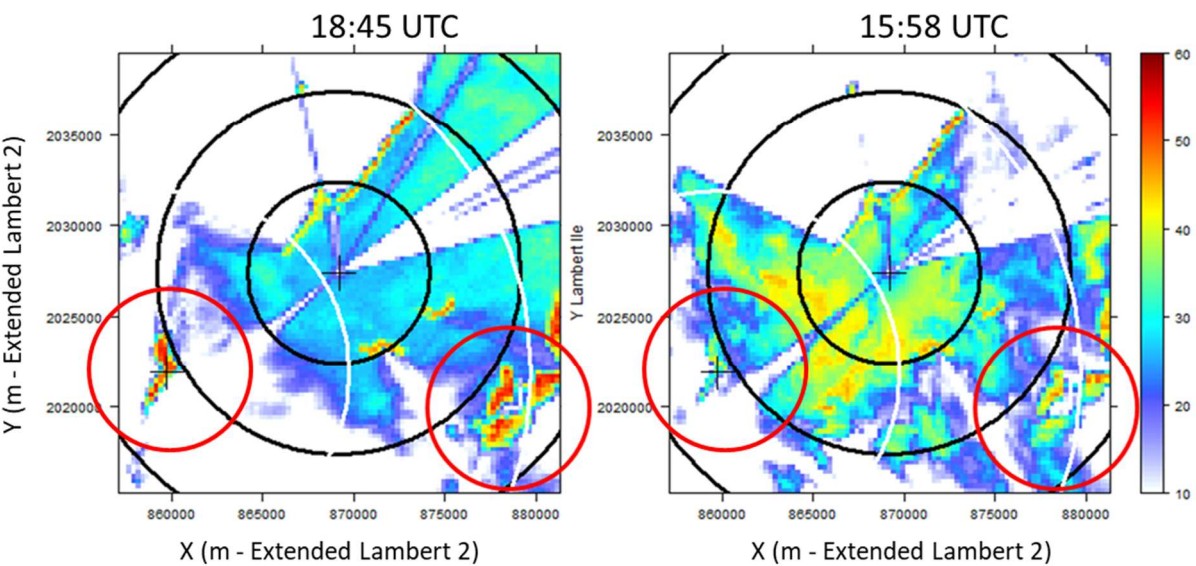

**Figure 3. Examples of XPORT 7.5° PPIs of raw reflectivity (not-corrected for attenuation) taken for two time steps during**
**the July 21, 2017 convective event. The crosses indicate the location of the two radars and the black / white (5 / 10-km)**
    **range markers correspond to the XPORT and the MOUC radar, respectively. The red circles focus the attention of the**
    **reader on the mountain returns associated with the Chamrousse (south-east) and the Moucherotte (south-west) mountains**
    **in between the 10-15 km range.**

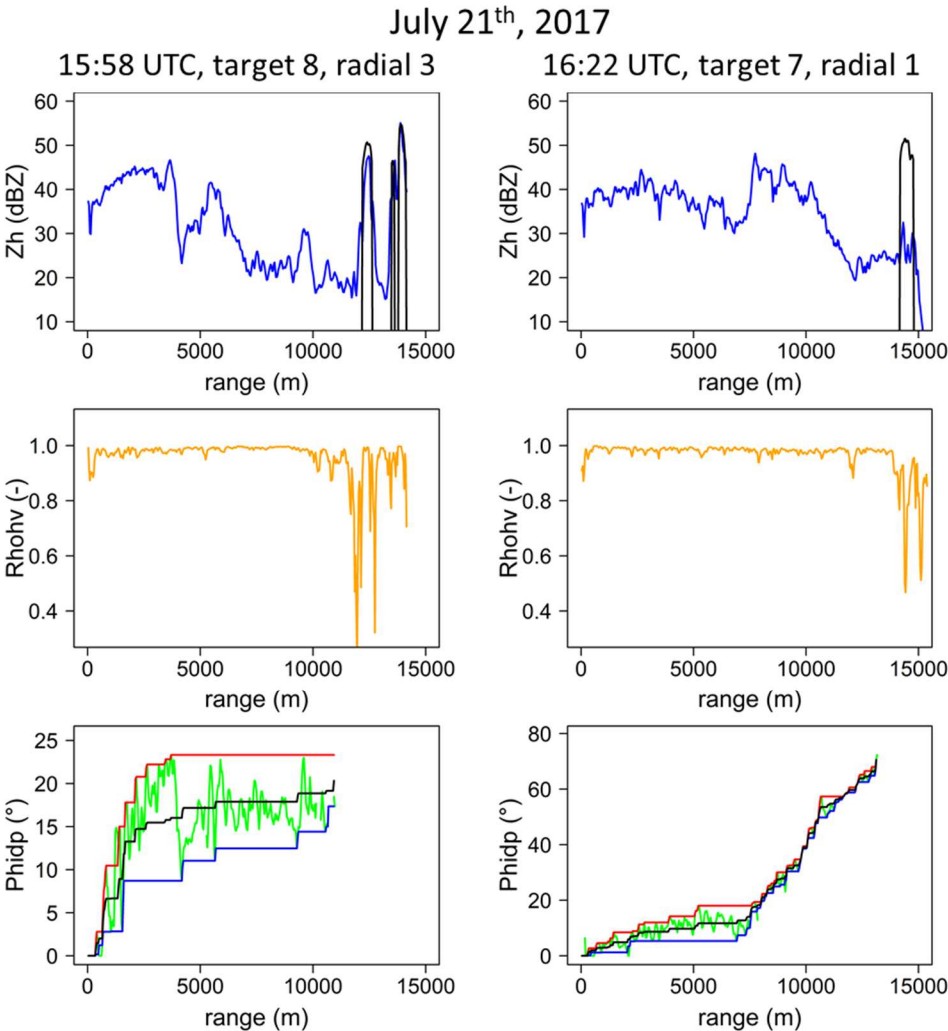

**Figure 4. Two examples (left, right) of $Z_h$, $\rho_{hv}$ and $\phi_{dp}$ range profiles of the XPORT radar (7.5°-PPI) during the July 21st, 2017 convective event for one radial of a given mountain target. The raw horizontal reflectivity profiles (top graphs) at the**
 **considered time steps (blue) are displayed together with the dry-weather reference target value (black). The $\rho_{hv}$ profiles (middle graphs) are used to detect the rainy gates not affected by clutter at close range and in the region of the mountain target. The bottom graphs display the raw $\psi_{dp}$ profiles (green), the upper (red) and lower (blue) envelope curves and the regularized $\phi_{dp}$ profiles (black).**

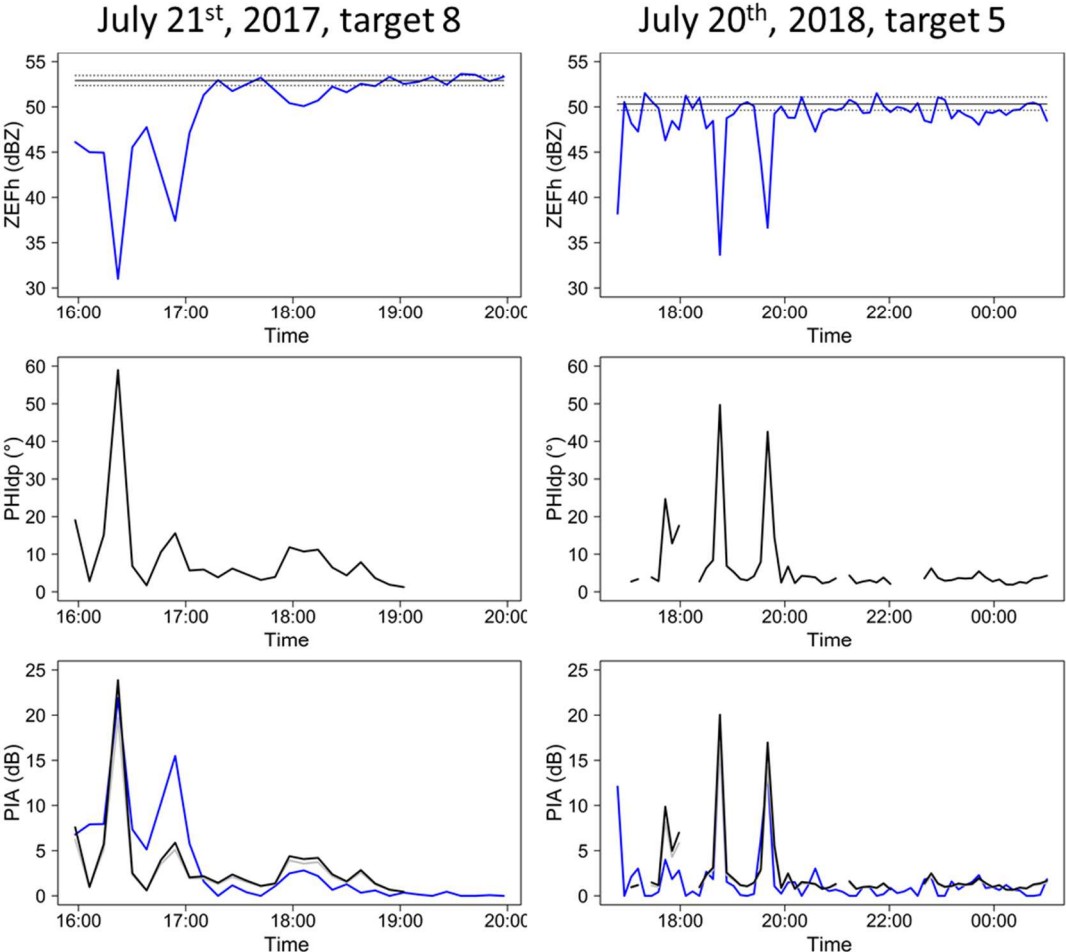

**Figure 5. Two examples of time series of the apparent reflectivity of mountain returns for a given target (top graphs), the**
**corresponding $\Phi_{dp}$ estimates (middle graphs) and the resulting PIA estimates. The horizontal black lines on top graphs**
**represent the mean (solid line), the 10% and 90% quantiles (dotted lines) of the dry-weather apparent reflectivity of the**
**target. The three lines on bottom graphs correspond to the MRT PIA estimate (blue) and to the polarimetry-derived PIA**
**estimates by using the linear $k - K_{dp}$ relationship (grey) and the non-linear $k - K_{dp}$ relationship (black), derived from**
**DSD measurements at ground level.**

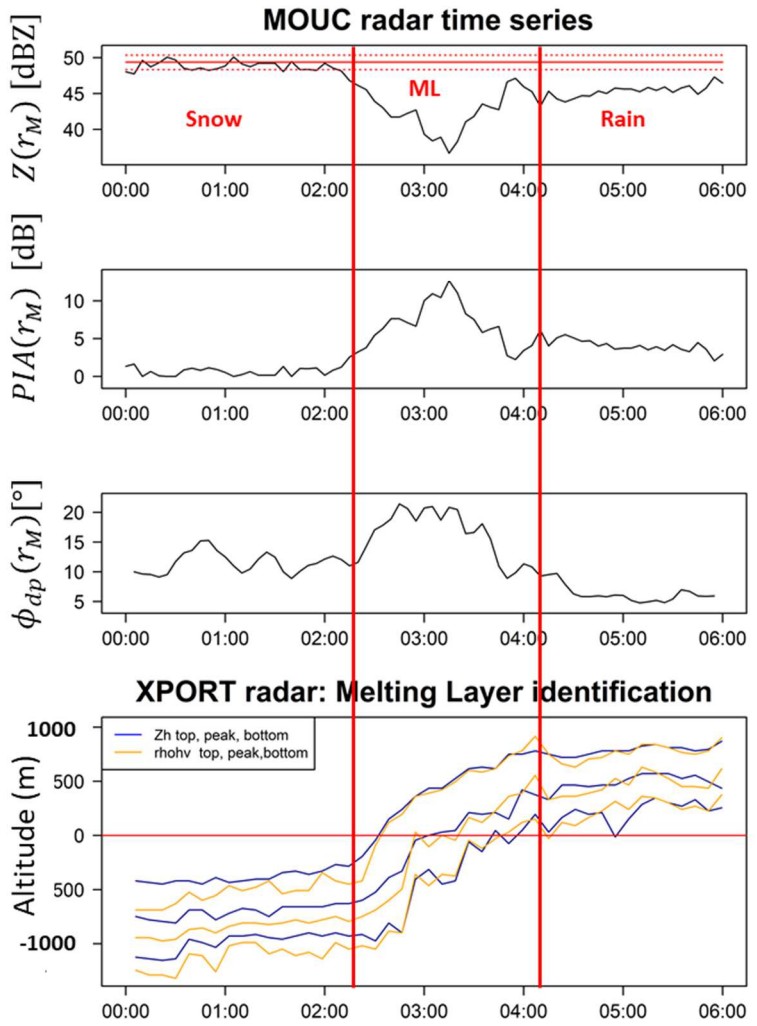

**Figure 6. Time series of (i) the apparent reflectivity values of a given mountain reference target together with the dry-weather reference (red horizontal lines for the mean (solid) and the 10% and 90% quantiles (dotted) , (ii) the resulting PIA estimates (dB), (iii) the corresponding $\phi_{dp}(r_M)$ values (°) for the 0°-PPI of the MOUC radar during the January 3-4th, 2018 stratiform rain event. The bottom graph displays the results of the ML detection algorithm performed with the XPORT 25°-PPI data; see text for details.**

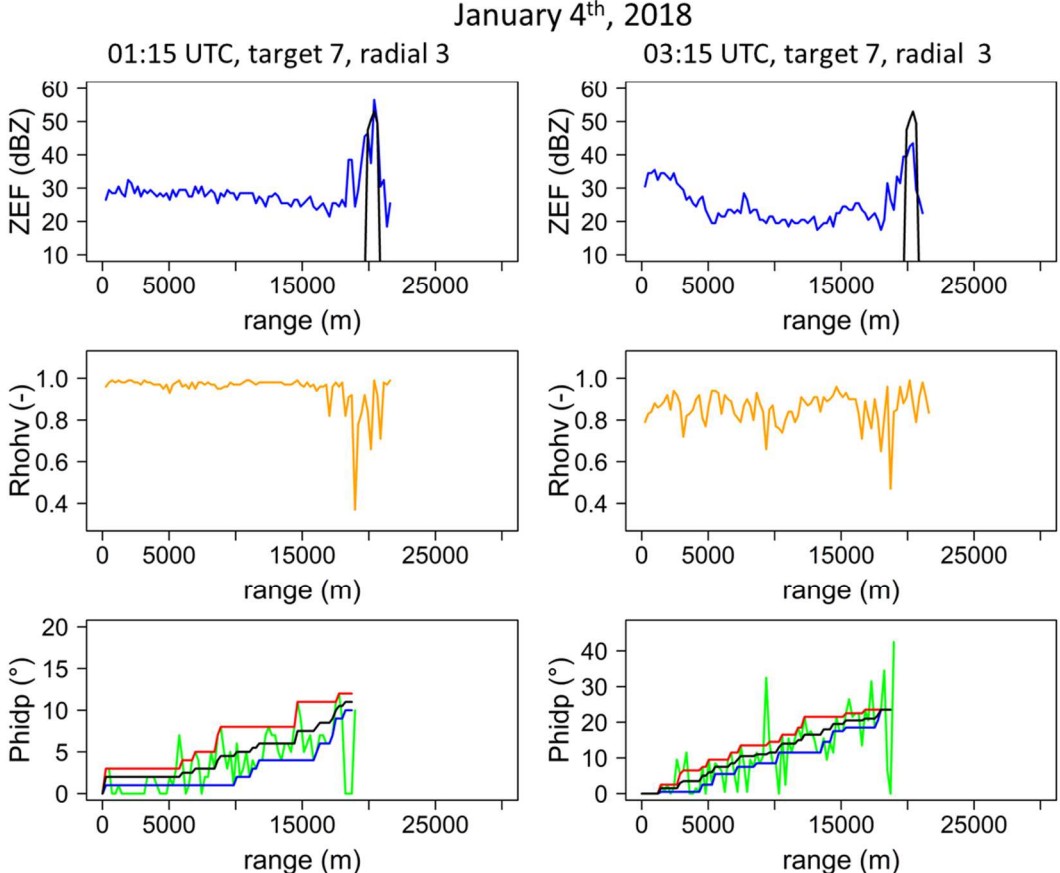

**Figure 7. Two examples (left, right) of $Z_h$, $\rho_{hv}$ and $\phi_{dp}$ range profiles of the MOUC radar (0°-PPI) during the July 21$^{st}$, 2017 convective event for one radial of a given mountain target. The raw horizontal reflectivity profiles (top graphs) at the considered time steps (blue) are displayed together with the dry-weather reference target value (black). The $\rho_{hv}$ profiles (middle graphs) are used to detect the rainy gates not affected by clutter at close range and in the region of the mountain target. The bottom graphs display the raw $\psi_{dp}$ profiles (green), the upper (red) and lower (blue) envelope curves and the**
 **regularized $\phi_{dp}$ profiles (black).**

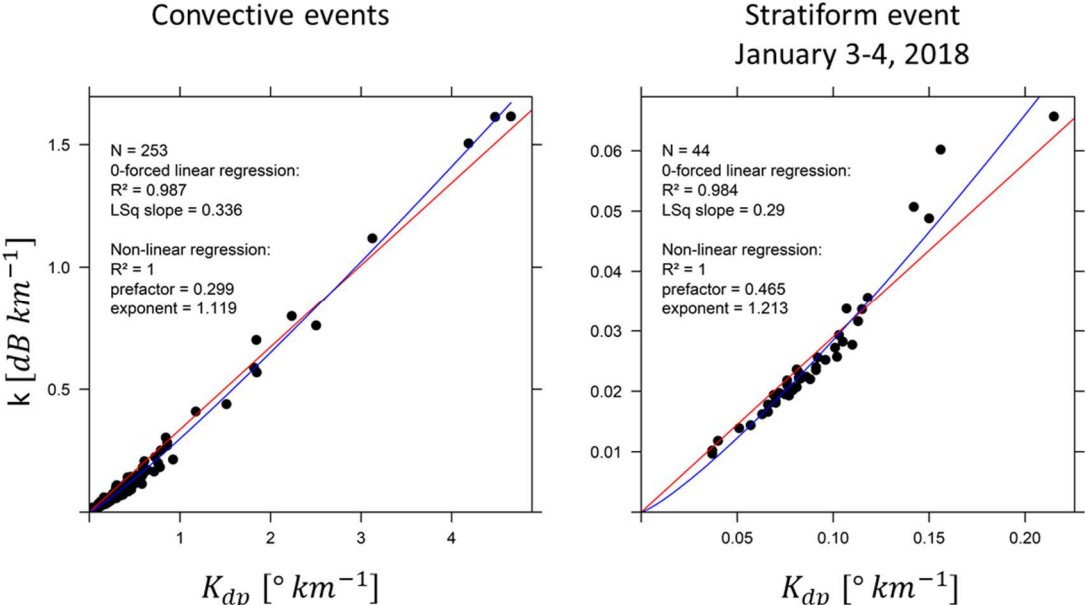

**Figure 8.** DSD-derived $k - K_{dp}$ relationships for the nine convective events (left) and for the stratiform event of January 3-4th, 2018; see text for details.


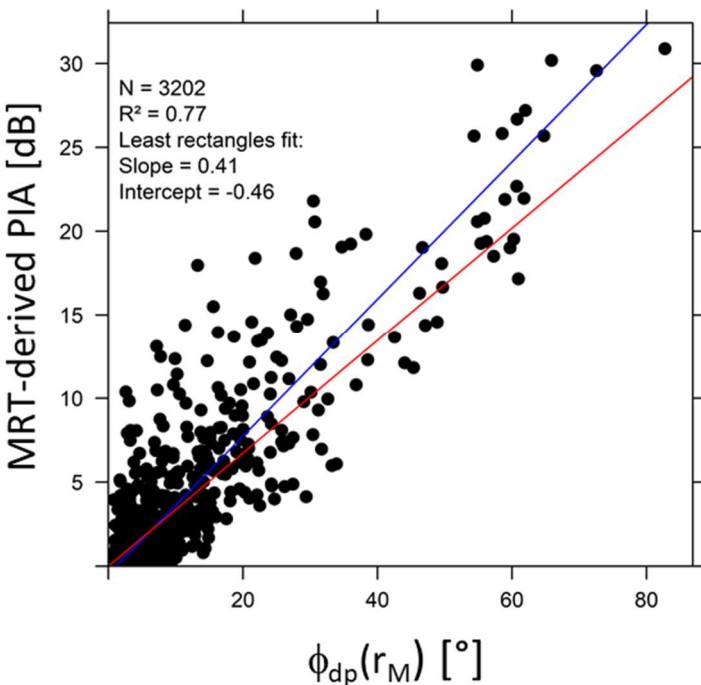

**Figure 9.** $PIA - \phi_{dp}$ **scatterplot for the nine convective events considered in this study. The blue line corresponds to the least rectangle fit to the data, while the red line corresponds to the linear** $k - K_{dp}$ **relationship derived from the DSD data available at**
**ground level.**


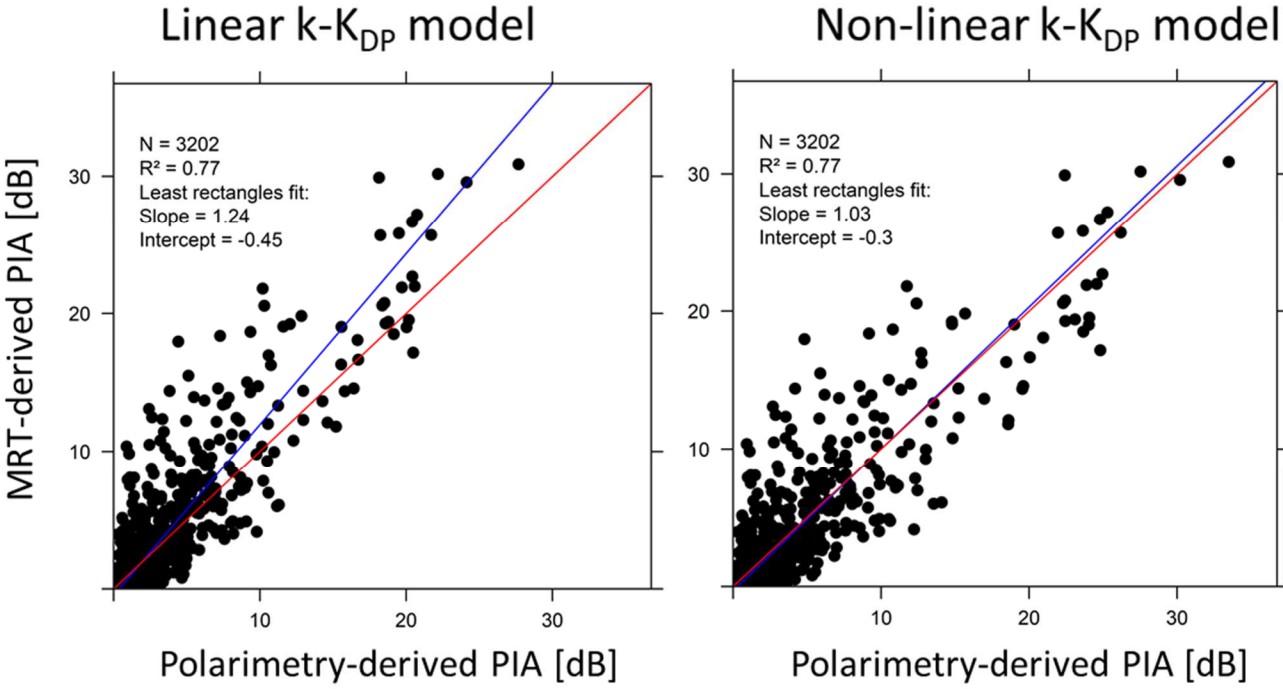


**Figure 10. Comparison of the PIAs derived from the Mountain Reference Technique and from polarimetry using the linear $k$-$K_{dp}$ relationship (left) and the non-linear $k$-$K_{dp}$ relationship (right) for the nine convective events. The blue line corresponds to the least-rectangle fit to the data and the red line is the 1/1 line.**


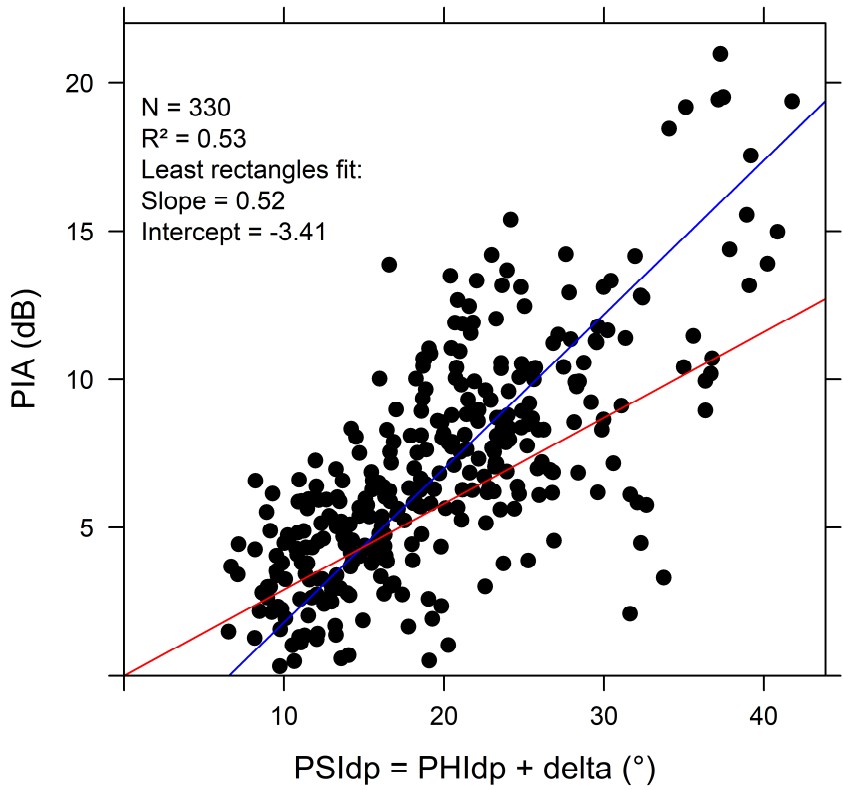

**Figure 11.** *PIA − ϕ$_{dp}$* scatterplot in the ML for the stratiform event of January 3-4[th], 2018. The blue line corresponds to the least rectangle fit to the data, while the red line corresponds to the linear $k − K_{dp}$ relationship derived from the DSD data available at ground level.



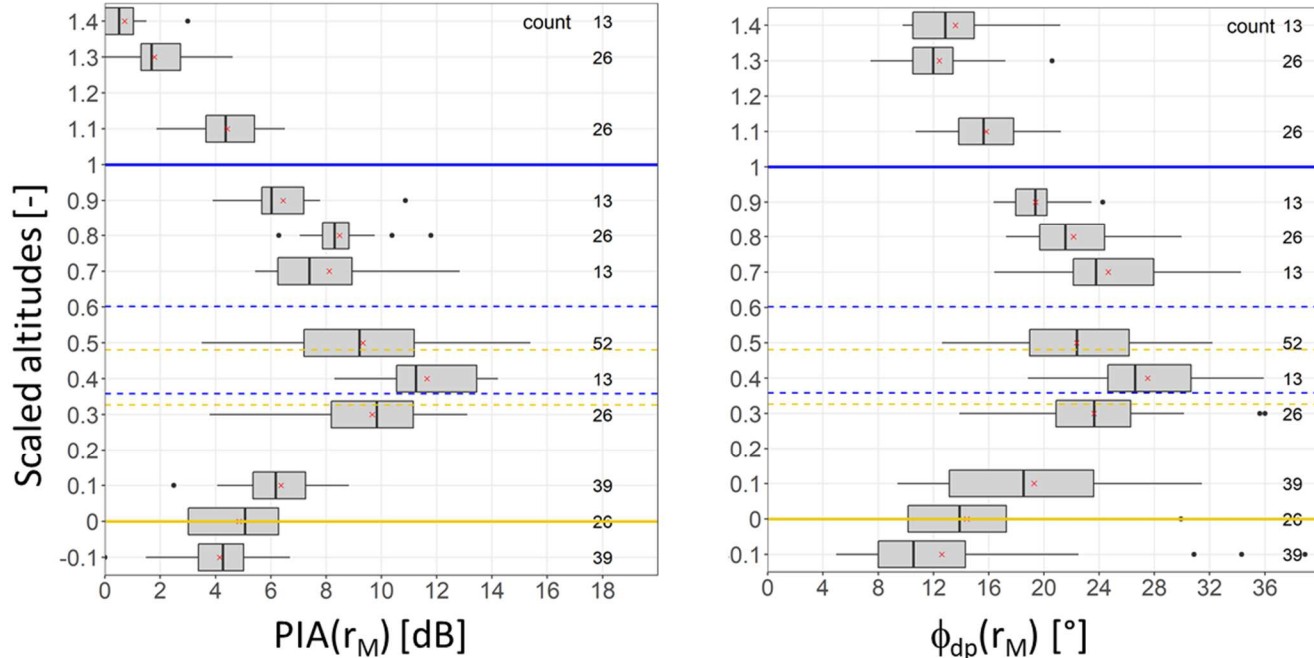


Figure 12. Box-plots of the *PIA* and $\phi_{dp}$ values within the ML as a function of the scaled altitude (left and right, respectively) for the stratiform event of January 4th, 2018. The horizontal blue and orange continuous lines represent the ML top and bottom, respectively; the dotted horizontal blue and orange lines give the 10 and 90% quantiles of the scaled altitudes of the Zh and $\rho_{hv}$

peak distributions, respectively.

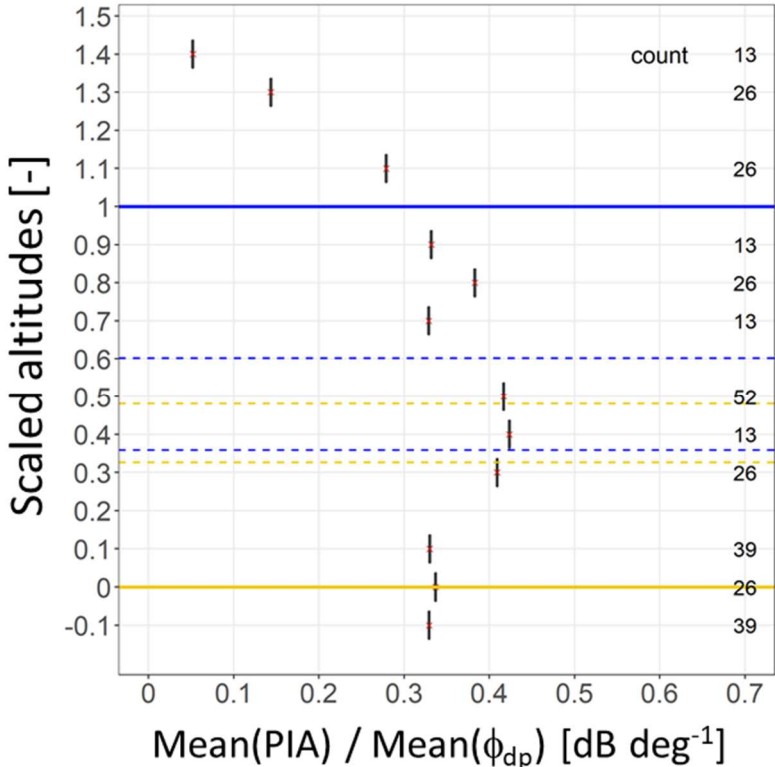

**Figure 13. Evolution of the ratio of the mean $PIA$ over the mean $\psi_{dp}$ values within the ML as a function of the scaled altitudes for the stratiform event of January 3-4th, 2018. The horizontal blue and orange lines represent the ML top and bottom, respectively; the dotted horizontal blue and orange lines give the 10 and 90% quantiles of the scaled altitudes of the Zh and $\rho_{hv}$ peak distributions, respectively.**