# Peer review of "Preliminary investigation of the relationship between differential phase shift and path-integrated attenuation at X-band in an Alpine environment"

_Atmospheric Measurement Techniques, 2019_

## Referee Comment (RC1) · Anonymous Referee #1 · 2 Mar 2020

**General comment**

The manuscript entitled "On the relationship between total differential phase and path integrated attenuation at X-band in an Alpine environment" presents interesting observations of radar measurements conducted at various relative altitudes with respect to the melting layer. The two-radar set-up and the combinations of their measurements is interesting, uncommon, and surely relevant for the radar meteorology community. I believe that the manuscript is suitable for publication after a major review, following the major and minor comments proposed here.

[Figure]

**Major comments**

1. Let us take as example Figure 4, but this has to be considered as a general comment on how to present the MRT data. When the authors show the reference dry value of reflectivity, i believe they should show also an indication of its variability (standard deviation or quantiles, to put some sort of error bars to the black curve). In my experience, the variability of mountain returns can be significant even at short time scales. This is particularly true as the radar of this manuscript is scanning and not pointing at a fixed direction. I would be pleased to see a significant section of the manuscript devoted to illustrate and statistically characterize the stability of MRT signals in dry weather before to discuss the analysis and the results of the two cases.

2. It would be beneficial if the authors could extend their analysis beyond the focus on two contrasted events only. It would be also more consistent with the title of the manuscript, that suggests a more global approach rather than the analysis of individual precipitation events.

3. While I found the data shown here very interesting, I could not see in the manuscript a clear research goal but rather a showcase of interesting radar observations.

**Other comments**

1. Abstract: I believe that the goals of this research should be better stated in the abstract.

2. Page 2, L 53: to my knowledge, the Swiss meteorology office has all the radars

installed at high altitude, i.e. it copes with the altitude dilemma by choosing visibility over proximity to the ground. Is it right?

3. Page 6, L 173: please consider that in case of hail of cm size, $\delta$ can be very large at X-band.

4. Page 7, L 200: the clutter identification by means of $\rho_{HV}$ should be interpreted as visual, or an algorithm is implemented to discriminate clutter from $\rho_{HV}$?

5. Page 7, L 191: was this choice based on comparison with ground-based instruments?

6. Page 6-7: is $K_{dp}$ then simply estimated as gate-by-gate derivative from the clean $\Phi_{dp}$, or an estimation method is used?

7. Page 11, L 345: would it be possible to show the position of the 16 MRT targets on a map? Also, could it be clarified more in detail how those (gates?pixels?) have been chosen, and which are their statistical properties?

8. Section 4.2: this one is in my opinion the most interesting part of the manuscript. I would recommend to expand it, and to apply this methodology to many more precipitation events and aim at results based on a large dataset.

9. Figure 4, please show all the polarimetric variables over the same range. For example the $\Psi_{dp}$ profile is shorter than the $Z_H$ or $\rho_{HV}$ profile. If a censoring is applied, please mention it in the caption and describe it in the text.

10. Figure 5: please mind the overlapping labels on the y axis.

---

## Referee Comment (RC2) · Anonymous Referee #2 · 2 Mar 2020

Review of the manuscript: On the relationship between total differential phase and path-integrated attenuation at X-band in an Alpine environment

By Delrieu et al.,

The manuscript discusses a methodology to investigate the relationship of the radar-derived PIA and the total differential phase in two different interesting precipitation regimes: rain and melting layer. I found the manuscript very well written and understandable and technically correct.

That said, I feel that the manuscript lack of significant conclusions. I suggest for major revision.

**Main concerns.**

1. The main messages to keep home for a reader seems to be i) apply a non-linear fit for k-Kdp relationship in rain to have an more unbiased estimation of PIA and ii) Melting layer attenuation can be estimated using a unique configuration that foresees the use of two radars optimally positioned in a Mountain environment. I find the fist finding not very new although useful, whereas I find the second finding interesting although the measurement configuration is far to be generalizable. I think the Authors should add some more text where they discuss they results thinking to a practical-oriented use of their findings. For example, keeping in mind all the limitations recalled by the Authors, do you encourage the use of the parametrization introduced in figure 9 (blue curve) to a have a rough estimation of ML attenuation using a polarimetric radar?
2. I was surprised by the fact that having two radars operating at nearly the same frequency in a such interesting configuration, somehow one above and one below the ML, you didn't try to compare the reflectivity factors of the two to have a proxy of the ML attenuation.
3. Did you check the radar absolute calibration using DSD Parsivel data?
4. MRT variability is never discussed in this manuscript. Do you think it can explain part of the variability in figure 8, y-axis?

---

## Referee Comment (RC3) · Anonymous Referee #3 · 4 Mar 2020

**1   Summary**

This manuscript proposes an data-driven investigation of the relationship between the differential phase shift and the specific attenuation in rain, melting snow and snow, using an original instrumental set-up consisting of two X-band polarimetric radars at different altitudes in the complex terrain around an Alpine valley. Such relationships are crucial to accurately correct for attenuation in precipitation to obtain reliable quantitative precipitation estimates at X-band.

The path integrated attenuation is determined using strong (fixed) mountains echoes

at various distances from the considered radar and provide independent estimates that can be compared to the (total) differential phase shift derived from polarimetric radar measurements. In rain, additional information about the raindrop size distribution measured by a disdrometer at the ground level is available to compute theoretical relationships. Focusing on two contrasted event (one convective and the other with a transition from snow to rain), the authors quantify the respective values of PIA and total differential phase shift from a number of mountains echoes, in rain using the lower radar, and in snow and in the melting layer using the higher radar. In this way, the specific attenuation in the ML can be quantified and it appears that the relationship between the PIA and the total differential phase shift is not that linear.

**2  Recommendation**

The manuscript is clear, the methods are sound and properly described. Such characterization of the attenuation in the melting layer and its links with the differential phase shift are relevant to the weather radar community and to AMT readership. I have some concerns and suggestions listed below, I hence recommend to send the manuscript back to the authors for major revisions.

**3  General comments**

1. The main concern in my view is the limited amount of data analyzed. The representativity of these two events, and the one used to investigate attenuation and differential phase shift in the melting layer, is not clearly addressed: to what extent can a reader use the numbers provided here for other locations/seasons? This is an important aspect because if not representative, the obtained results will be of limited interest to potential readers (who may not be able to reproduce the same

instrumental set-up involving two radars and complex terrain). The authors touch upon this issue in the conclusions and mention that they will process more data, but this should be addressed earlier in the text, and to be honest I am wondering if they should not do so already in this manuscript.

2. The scientific objectives of the manuscript are not very clear. What are the main take-home messages for the reader?

3. The assumption that the differential phase shift on backscatter ($\delta_{hv}$) is negligible is not really justified. Together with the possible PIA overestimation due to radome attenuation for the MOUC radar during the stratiform event, these two sources of uncertainties may affect the highlighted behavior of the ratio between the PIA and the $\Psi_{dp}$ in the ML. This aspect should be clarified.

**4  Specific comments**

1. Title: I think the exact term is differential phase **shift**. I recommend the authors to edit the whole text to add shift where needed.

2. P.1, l.12: rainfall and snowfall rather than rain and snow.

3. P.1, l.13: "high mountain regions": the adjective high is relative... I suggest to change to "mountainous regions".

4. P.1, l.24: high rather than strong rain rates.

5. P.1, l.24: $\Phi_{dp}$ is not defined yet.

6. P.2, l.41: insert "over extended areas" between "achieved" and "with traditional".

[Figure]

7. P.2, l.42-51: it would be good to support the statements by references to the literature.

8. P.2, l.58: the common usage is that polarimetric means dual-polarization and Doppler...

9. P.3, l.72: $K_{dp}$is the specific differential phase shift on propagation. Please correct wherever needed in the text.

10. P.4, Section 2.1: what about the calibration of the two radars? How was it checked/performed?

11. P.4, l.110: missing closing bracket after "study".

12. P.5, Eq.1: This equation is for a given polarization, this should be indicated using a subscript $h/v$ for instance.

13. P.6, l.168: $\delta_{hv}$ is the differential phase shift on backscatter.

14. P.6, l.175: the units of these ranges of values (degree?) should be provided.

15. P.6, l.179: the assumption of negligible $\delta_{hv}$ should be better justified. A few degrees for $\delta_{hv}$ as suggested on l.175 are not necessarily negligible compared to the overall $\Psi_{dp}$ values provided in Fig.10 for instance. As mentioned in the General Comments, the resulting uncertainty in $\Phi_{dp}$ values may affect the behavior highlighted in Fig.10 and 11. Combined with possible radome attenuation...

16. P.6, l.182-183: why $N = 10$ and $N = 4$? How did you come up with these values?

17. P.7, l.191: same here, please justify these thresholds in $Z_h$ and $\rho_{hv}$.

18. P.7, l.196: the black line in Fig.4 represents the instantaneous values of $Z_h$, it would be nice to figure the variability of the mountain return, to give the reader an

idea about the noise of such echoes (and hence an idea about the uncertainty in the derived PIA estimates).

19. P.8, l.218: please provide a reference for negligible attenuation in snow.

20. P.8, l.228: "[7]" seems to be a literature reference, but there is no number in the references. Please update.

21. P.8, l.245: the co-fluctuation between the two signals does not look that bad by eye... Maybe you could compute the correlation coefficient to have a quantitative criterion?

22. P.8, l.242-248: the possible influence of beam broadening and radome attenuation (see l.401-406) could be first mentioned here.

23. P.9, l.277: change citations from numbers ([10] and [19]) to author's names...

24. P.10, l.302 and Fig.8: I may be wrong, but I think there is an issue with the axis labels in Fig.8: PIA from polarimetry should be on the y axis while the PIA from MRT should be on the x-axis. Otherwise, there would be an underestimation from the polarimetric approach (slope $>$ 1), not consistent with Fig.6 left. Please clarify.

25. P.11, l.322-323: what can explain this variability in the ML depth? If this is due to different types of hydrometeors, is the scaling approach used here still relevant?

26. P.11, l.342: could this less evident shift between peak in $Z_h$ and in $\rho_{hv}$ be also due to beam broadening? As the ML is going up in altitude, it is also going further away in the PPI used to extract the polarimetric radar variables...

27. P.12, l.379: why are $\delta_{hv}$ values expressed in dB?

28. P.13, l.405-406: but the attenuation due to wet snow sticking on the radome is not necessarily directly proportional to the rain rate (it can accumulates...). The assumption of negligible radome attenuation during the ML scans should be better justified. As it could have significant impact on the estimated PIA values and hence on the behavior of the ratio PIA/$\Psi_{dp}$ in Fig.11.

29. P.17, Table 1: the spectral width is not recorded?

30. P.19, Fig.2: it would be better to use the same y axis scale between the 2 events, to ease the comparison.

31. P.20, Fig.3: the underlying images are too coarse in resolution. They should be improved.

32. P.21, Fig.4: As expected, the phase measurements are contaminated by clutter earlier (i.e. closer to the radar) than reflectivity measurements. Hence the last (starting from the radar) reliable gate in $\Psi_{dp}$ may be closer to the radar than the last reliable gate in $Z_h$ from which the PIA is estimated. Could this introduce a bias?

---

## Author Comment (AC1) · 22 Apr 2020

First, we would like to thank the three reviewers for their detailed and constructive comments about this article.

As a main feedback, the three reviewers suggest us to consider more events in our analyses in order to strengthen the quantitative outcomes of this article, especially regarding the PIA-dp relationship in the melting layer (ML). We recognise this is desirable and feasible since we have recorded about 30 events with the ML being at the level of the Moucherotte radar. This will be however a major additional work requiring the collection and processing of the Moucherotte radar data that are not available yet to

us, and will not be available in the coming weeks/months due to the covid-19 lockdown. However, we have extended the rain case study to 9 convective events (new Table 2) and the results obtained nicely confirm the analysis of the first version of the article for the July 21st 2017 convective event.

We have also deepened the methodology and redo all the calculations with a special attention on (i) the characterization of the dry-weather reference targets stability and time variability (new tables 3 and 4, figures modification to show the 10% and 90% quantiles of the apparent reflectivity of the mountain targets) and (ii) on the possible deltahv contamination of the raw psidp profiles. The regularization procedure of the raw phidp profiles was improved in this latter respect and we found it to be efficient in filtering "bumps" likely associated with deltahv contamination. Regarding the manuscript, the abstract and the conclusion were largely rewritten and the description of the MRT and polarimetry PIA estimators was also much detailed in sections 3.1 and 3.2. Two additional figures were included to better illustrate and support the analyses made. In general terms, we took great care in discussing the results and the possible influence of the various sources of error in the two different case studies. We do hope these efforts, which effectively resulted in a major revision, will satisfy the anonymous reviewers.

Our item-by-item replies are inserted below in blue within the reviewers' comments recalled in black.

Anonymous Referee #1

General comment The manuscript entitled "On the relationship between total differential phase and path integrated attenuation at X-band in an Alpine environment" presents interesting observations of radar measurements conducted at various relative altitudes with respect to the melting layer. The two-radar set-up and the combinations of their measurements is interesting, uncommon, and surely relevant for the radar meteorology community. I believe that the manuscript is suitable for publication after a major review, following the major and minor comments proposed here. Major comments 1.

Let us take as example Figure 4, but this has to be considered as a general comment on how to present the MRT data. When the authors show the reference dry value of reflectivity, I believe they should show also an indication of its variability (standard deviation or quantiles, to put some sort of error bars to the black curve). In my experience, the variability of mountain returns can be significant even at short time scales. This is particularly true as the radar of this manuscript is scanning and not pointing at a fixed direction. I would be pleased to see a significant section of the manuscript devoted to illustrate and statistically characterize the stability of MRT signals in dry weather before to discuss the analysis and the results of the two cases.

A considerable effort was done in this respect with a detailed description and illustration of the methods used to select the mountain targets and their stability and time variability in the two measurement configurations (new section 3.1). Two tables were added and the figures were modified when needed so as to show the time variability of the mountain targets. We did not modify Fig.4 however for which the considered mountain returns correspond only to those available for a given radial of the considered target.

2. It would be beneficial if the authors could extend their analysis beyond the focus on two contrasted events only. It would be also more consistent with the title of the manuscript, that suggests a more global approach rather than the analysis of individual precipitation events.

As indicated above we have extended the convective case study to 9 events. We are not in position to do the same work for the ML case study in this period. We have moderated the ambition of our study by adding "Preliminary investigation of . . ." in the title of the article

3. While I found the data shown here very interesting, I could not see in the manuscript a clear research goal but rather a showcase of interesting radar observations.

We tried to improve the motivation of this study in several places. Effectively, this study is somewhat "upstream" with respect to the practical goals of the RadAlp experiment

which concern rainfall and snowfall estimation in a high-mountain context. This is only one step in a certainly long-term process. The reviewer may recognised the importance of comparing PIAs derived from polarimetry and from direct power estimates and the need to put these 2 estimators in competition for QPE with respect to independent measurements (future step). Attenuation in the ML is also poorly documented and important for the interpretation of radar measurements, especially in our high-mountain context. Other comments 1. Abstract: I believe that the goals of this research should be better stated in the abstract.

Modification performed: "We present in this article a methodology for studying the relationship between the differential phase shift due to propagation in precipitation ($\Phi$_dp) and path-integrated attenuation (PIA) at X-Band. This relationship is critical for quantitative precipitation estimation (QPE) based on polarimetry due to severe attenuation effects in rain at the considered frequency. In addition, this relationship is still poorly documented in the melting layer (ML) due to the complexity of the hydrometeors' distributions in terms of size, shape and density. The available observation system offers promising features to improve this understanding and to subsequently better process the radar observations in the ML."

2. Page 2, L 53: to my knowledge, the Swiss meteorology ofiňĄce has all the radars installed at high altitude, i.e. it copes with the altitude dilemma by choosing visibility over proximity to the ground. Is it right?

Yes, and this is the same for the French radar network. This is justified for the detection/monitoring of strong and localized convective events at the regional scale that are poorly sensed by conventional raingauge networks. However, we showed in a previous article that the Moucherotte radar perfoms its measurements within or above the ML in about 70% of cases of significant precipitation in Grenoble, with subsequent increased difficulties for QPE at ground level. See the following reference: A.K. KHANAL, G. DELRIEU, F. CAZENAVE and B. BOUDEVILLAIN, 2019. Radar Remote Sensing of Precipitation in High Mountains: Detection and Characterization of Melting Layer in the

Grenoble valley, French Alps, Atmosphere, 10, 784; doi:10.3390/atmos10120784

3. Page 6, L 173: please consider that in case of hail of cm size, $\delta$ can be very large at X-band.

Yes we are aware of this. No hail was reported for the convective cases considered. In addition we took great care in the revision to evidence and try to filter (with some success) the deltahv contaminations, e.g. new Fig.4 left.

4. Page 7, L 200: the clutter identification by means of HV should be interpreted as visual, or an algorithm is implemented to discriminate clutter from HV ?

See next point

5. Page 7, L 191: was this choice based on comparison with ground-based instruments?

No this choice is based on the ML statistics presented in Khanal et al. (2019) As detailed in new section 3.2, we flagged as noise all $\Phi_{(dp)}$gates for which _hv < 0.95 for the XPORT rain case study and we determined the beginning and the end of the precipitation range considering a number of successive gates (10, i.e. e range extent of 342 m) for which this threshold was overpassed. Due to the noise affecting rhohv in the ML, we had to considerably lower this threshold (down to 0.8) for the ML case study. We have abandoned the idea of using a threshold on the reflectivity in the new version of our methodology

6. Page 6-7: is Kdp then simply estimated as gate-by-gate derivative from the clean $\Phi$dp, or an estimation method is used?

Yes, the Kdp profile is simply estimated as gate-by-gate derivative of the processed $\Phi_{dp}$profile.

7. Page 11, L 345: would it be possible to show the position of the 16 MRT targets on a map? Also, could it be clarified more in detail how those (gates? pixels?) have been

chosen, and which are their statistical properties?

The way the targets are defined is now more detailed in the revision in section 3.1, with 2 additional tables and modification of the relevant figures

8. Section 4.2: this one is in my opinion the most interesting part of the manuscript. I would recommend to expand it, and to apply this methodology to many more precipitation events and aim at results based on a large dataset.

More details have been added to illustrate the methodology and the limitations of this preliminary case study. As explained in the head of the review, we are unfortunately not in position to extend it to other events right now...

9. Figure 4, please show all the polarimetric variables over the same range. For example the $\Psi dp$ profile is shorter than the ZH or HV profile. If a censoring is applied, please mention it in the caption and describe it in the text.

The $\Phi\_dp$ profile processing / display is voluntarily restricted to the "rainy" region (r0, rM) free of close-range and mountain clutter while the other profiles include the mountain target. 10. Figure 5: please mind the overlapping labels on the y axis. corrected

---

## Author Comment (AC2) · 22 Apr 2020

First, we would like to thank the three reviewers for their detailed and constructive comments about this article.

As a main feedback, the three reviewers suggest us to consider more events in our analyses in order to strengthen the quantitative outcomes of this article, especially regarding the PIA-dp relationship in the melting layer (ML). We recognise this is desirable and feasible since we have recorded about 30 events with the ML being at the level of the Moucherotte radar. This will be however a major additional work requiring the collection and processing of the Moucherotte radar data that are not available yet to

us, and will not be available in the coming weeks/months due to the covid-19 lockdown. However, we have extended the rain case study to 9 convective events (new Table 2) and the results obtained nicely confirm the analysis of the first version of the article for the July 21st 2017 convective event.

We have also deepened the methodology and redo all the calculations with a special attention on (i) the characterization of the dry-weather reference targets stability and time variability (new tables 3 and 4, figures modification to show the 10% and 90% quantiles of the apparent reflectivity of the mountain targets) and (ii) on the possible deltahv contamination of the raw psidp profiles. The regularization procedure of the raw phidp profiles was improved in this latter respect and we found it to be efficient in filtering "bumps" likely associated with deltahv contamination. Regarding the manuscript, the abstract and the conclusion were largely rewritten and the description of the MRT and polarimetry PIA estimators was also much detailed in sections 3.1 and 3.2. Two additional figures were included to better illustrate and support the analyses made. In general terms, we took great care in discussing the results and the possible influence of the various sources of error in the two different case studies. We do hope these efforts, which effectively resulted in a major revision, will satisfy the anonymous reviewers. Our item-by-item replies are inserted below in blue within the reviewers' comments recalled in black.

Anonymous Referee #2 The manuscript discusses a methodology to investigate the relationship of the radar-derived PIA and the total differential phase in two different interesting precipitation regimes: rain and melting layer. I found the manuscript very well written and understandable and technically correct. That said, I feel that the manuscript lack of significant conclusions. I suggest for major revision. Main concerns.

1. The main messages to keep home for a reader seems to be i) apply a non-linear fit for k-Kdp relationship in rain to have an more unbiased estimation of PIA and ii) Melting layer attenuation can be estimated using a unique configuration that foresees the use of two radars optimally positioned in a Mountain environment. I find the first

finding not very new although useful, Yes but several publications (e.g. Testud et al. 2000, Schneebeli and Berne 2012) mention the existence of a linear relationship at X-band and the subsequent advantages in terms of QPE. Our preliminary findings seem to indicate that this is not the case and that the rain type may be an important factor controlling this relationship. whereas I find the second finding interesting although the measurement configuration is far to be generalizable. I think the Authors should add some more text where they discuss their results thinking to a practical-oriented use of their findings. For example, keeping in mind all the limitations recalled by the Authors, do you encourage the use of the parametrization introduced in figure 9 (blue curve) to a have a rough estimation of ML attenuation using a polarimetric radar? No because in this scatterplot are mixed pairs of estimates obtained from various layers of the melting layer. In the hypothesis of a linear relationship, new Fig. 13 (old Fig 11) is certainly more useful to describe the k-Kdp relationship and its variation within the ML.

2. I was surprised by the fact that having two radars operating at nearly the same frequency in a such interesting configuration, somehow one above and one below the ML, you didn't try to compare the reflectivity factors of the two to have a proxy of the ML attenuation. This is a good idea that was already explored by our Météo-France co-authors for the other Alpine X-band radars (within the RythMME project). See the following reference: Yu, N., Gaussiat, N., and Tabary, P.: Polarimetric X‐band weather radars for quantitative precipitation estimation in mountainous regions. Q. J. Royal Meteorol. Soc., 144(717), DOI:10.1002/qj.3366, 2018. Note that an accurate calibration of the two radars is required and that the difference in the resolution volumes limit to some extent the interest of this approach. In addition, our colleague Nan Yu, started looking at the possibility to implement his method to the Grenoble configuration. He found out that many of the "common measurement gates" of the two radars were actually affected by side-lobe contamination for the MOUC radar.

3. Did you check the radar absolute calibration using DSD Parsivel data? The MOUC radar calibration was performed through the Météo-France standard electronic calibration procedure followed by a qualification of the rain products through radar-raingauge comparisons. The XPORT radar electronic calibration was performed at various occasions during the radar implementation in several campaigns. No radar-raingauge or radar-disdrometer comparisons have been made so far in the Grenoble context. In any case, a major advantage of the proposed methodology is that both the $\Phi\_dp$ and the MRT PIA estimates are independent of eventual radar calibration errors.

4. MRT variability is never discussed in this manuscript. Do you think it can explain part of the variability in figure 8, y-axis? As requested by reviewer #1, we have added a lot of material in the revision regarding the mountain return stability and time variability. For the rain case, we believe this factor to be of very limited importance due to the very small variability of the mountain returns. More impact is likely for the ML case but we don't think this is a dominant source of error.

---

## Author Comment (AC3) · 22 Apr 2020

First, we would like to thank the three reviewers for their detailed and constructive comments about this article.

As a main feedback, the three reviewers suggest us to consider more events in our analyses in order to strengthen the quantitative outcomes of this article, especially regarding the PIA-dp relationship in the melting layer (ML). We recognise this is desirable and feasible since we have recorded about 30 events with the ML being at the level of the Moucherotte radar. This will be however a major additional work requiring the collection and processing of the Moucherotte radar data that are not available yet to

us, and will not be available in the coming weeks/months due to the covid-19 lockdown. However, we have extended the rain case study to 9 convective events (new Table 2) and the results obtained nicely confirm the analysis of the first version of the article for the July 21st 2017 convective event.

We have also deepened the methodology and redo all the calculations with a special attention on (i) the characterization of the dry-weather reference targets stability and time variability (new tables 3 and 4, figures modification to show the 10% and 90% quantiles of the apparent reflectivity of the mountain targets) and (ii) on the possible deltahv contamination of the raw psidp profiles. The regularization procedure of the raw phidp profiles was improved in this latter respect and we found it to be efficient in filtering "bumps" likely associated with deltahv contamination. Regarding the manuscript, the abstract and the conclusion were largely rewritten and the description of the MRT and polarimetry PIA estimators was also much detailed in sections 3.1 and 3.2. Two additional figures were included to better illustrate and support the analyses made. In general terms, we took great care in discussing the results and the possible influence of the various sources of error in the two different case studies. We do hope these efforts, which effectively resulted in a major revision, will satisfy the anonymous reviewers.

Our item-by-item replies are inserted below in blue within the reviewers' comments recalled in black.

Anonymous Referee #3 1 Summary This manuscript proposes a data-driven investigation of the relationship between the differential phase shift and the speciffc attenuation in rain, melting snow and snow, using an original instrumental set-up consisting of two X-band polarimetric radars at different altitudes in the complex terrain around an Alpine valley. Such relationships are crucial to accurately correct for attenuation in precipitation to obtain reliable quantitative precipitation estimates at X-band. The path integrated attenuation is determined using strong (fixed) mountains echoes at various distances from the considered radar from the two considered radars. . . and provide independent estimates that can be compared to the (total) differential phase shift derived from polarimetric radar measurements. In rain, additional information about the raindrop size distribution measured by a disdrometer at the ground level is available to compute theoretical relationships. Focusing on two contrasted event (one convective and the other with a transition from snow to rain), the authors quantify the respective values of PIA and total differential phase shift from a number of mountains echoes, in rain using the lower radar, and in snow and in the melting layer using the higher radar. In this way, the specific attenuation in the ML can be quantified and it appears that the relationship between the PIA and the total differential phase shift is not that linear. To be more precise, in the ML (stratiform case), old Fig. 11 suggests that the multiplicative coefficient of a k-Kdp relationship (assumed to be linear) depends on the position within the ML and as such on the melting processes.

2 Recommendation The manuscript is clear, the methods are sound and properly described. Such characterization of the attenuation in the melting layer and its links with the differential phase shift are relevant to the weather radar community and to AMT readership. I have some concerns and suggestions listed below, I hence recommend to send the manuscript back to the authors for major revisions.

3 General comments 1. The main concern in my view is the limited amount of data analyzed. The representativity of these two events, and the one used to investigate attenuation and differential phase shift in the melting layer is not clearly addressed: to what extent can a reader use the numbers provided here for other locations/seasons? This is an important aspect because if not representative, the obtained results will be of limited interest to potential readers (who may not be able to reproduce the same instrumental set-up involving two radars and complex terrain). The authors touch upon this issue in the conclusions and mention that they will process more data, but this should be addressed earlier in the text, and to be honest I am wondering if they should not do so already in this manuscript. We fully understand this comment but we are not in position to extend our analyses to other ML cases right now, see our head comment. . . We have extended the convective case study to 9 events.

2. The scientific objectives of the manuscript are not very clear. What are the main take-home messages for the reader? We have tried to improve this aspect in the conclusion of the article (and in the abstract). Message 1 about the rain case study in the conclusion: "In the end, the scatterplot of the MRT PIAs as a function of the $\Phi\_dp$ ($r\_M$) for all the nine convective events presents an overall good coherence with however a significant dispersion (explained variance of 77%). It is interesting to note that the non-linear k- $K\_dp$ relationship derived from independent DSD measurements taken during the events of interest at ground level allows a satisfactory transformation of the XPORT $\Phi\_dp$ profiles into almost unbiased (although dispersed) PIA estimates. Both estimation methods are prone to specific errors and, even if the MRT PIA estimator is more directly related to power attenuation, it is a priori difficult to say which estimator is the best. An assessment exercise of attenuation correction algorithms, making use of both PIA estimators, with respect to an independent data source (e.g. raingauge measurements) is desirable to distinguish the two PIA estimators. A specific experiment is being designed in this perspective to be implemented in the near future." Message 2 about the ML case study:

"From this dataset, it was possible to derive the evolution of PIA($r\_M$) and $\Phi\_dp$ ($r\_M$) values as a function of the altitude within the ML. The evolution with the altitude of the ratio of the mean value of PIA($r\_M$) over the mean value of $\Phi\_dp$ ($r\_M$), as a proxy for the slope of a linear k- $K\_dp$ relationship within the ML, was also considered. . . The three variables considered present a clear signature as a function of the (scaled) altitude. In particular, the PIA/$\Phi\_dp$ ratio peaks at the level of the _hv peak (somewhat lower than the Zh peak), with a value of 0.42 dB degree-1, while its value in rain just below the ML is 0.33 dB degree-1. . . . Although the experimental configuration for the study of attenuation in the ML presents some limitations (radome attenuation, NUBF), the preliminary results presented here will be deepened by processing a dataset of about thirty stratiform events with the presence of the ML at the level of the MOUC radar.

3. The assumption that the differential phase shift on backscatter ($\delta$hv) is negligible is not really justified. This comment led us to a big consideration of this point during the revision process. Besides the cited literature we have searched evidence of deltahv contamination in the raw psidp profiles both for the convective cases and the stratifom one. We have found some "bumpy" profiles during some convective events that our regularization procedure is fortunately able to filter in a nice way. No bumpy profiles in the stratiform case. But we recognise that more work is required on this topic... Together with the possible PIA overestimation due to radome attenuation for the MOUC radar during the stratiform event, these two sources of uncertainties may affect the highlighted behavior of the ratio between the PIA and the $\Psi$dp in the ML. This aspect should be clarified. This is actually hard to clarify. In addition to the delthv contamination, the radome attenuation is a real concern. We have considered a mountain target in the vicinity of the MOUC radar (5 km or so) but it was too unstable to provide useful information. We have tried to be more careful in our interpretations of this very limited case study ; see our new comments of the results obtained on the upper part of the ML (which could sign radome attenuation or NUBF effects). Radome attenuation is likely small for the considered case due to the low rainrates/snowrates and the fact the radome is heated.

4 Specific comments 1. Title: I think the exact term is differential phase shift. I recommend the authors to edit the whole text to add shift where needed. OK, done

2. P.1, l.12: rainfall and snowfall rather than rain and snow. corrected

3. P.1, l.13: "high mountain regions": the adjective high is relative... I suggest to change to "mountainous regions". Although relative, we want to keep the adjective "high" since radar QPE is particularly challenging in such regions wrt to plains or medium-elevation mountains

4. P.1, l.24: high rather than strong rain rates. corrected

5. P.1, l.24: $\Phi$dp is not defined yet. corrected
6. P.2, l.41: insert "over extended areas" between "achieved" and "with traditional". done

7. P.2, l.42-51: it would be good to support the statements by references to the literature. These statements are quite "generic" and do not require in our view specific referencing.

8. P.2, l.58: the common usage is that polarimetric means dual-polarization and Doppler... Good point, we suppressed "Doppler"

9. P.3, l.72: Kdp is the specific differential phase shift on propagation. Please correct wherever needed in the text. corrected

10. P.4, Section 2.1: what about the calibration of the two radars? How was it checked/performed? The MOUC radar calibration was performed through the Météo-France standard electronic calibration procedure followed by a qualification of the rain products through radar-raingauge comparisons. The XPORT radar electronic calibration was performed at various occasions during the radar implementation in several campaigns. No radar-raingauge or radar-disdrometer comparisons have been made so far in the Grenoble context. In any case, a major advantage of the proposed methodology is that both the $\Phi\_dp$ and the MRT PIA estimates are independent of eventual radar calibration errors.

11. P.4, l.110: missing closing bracket after "study". corrected

12. P.5, Eq.1: This equation is for a given polarization, this should be indicated using a subscript h/v for instance. The following sentence was added: Note that PIAs can be obtained from eq.1 for both the horizontal and the vertical polarizations. In the present article, we will restrict ourselves to the horizontal polarization, the study of differential attenuation being a possible topic for a future study.

13. P.6, l.168: $\delta$hv is the differential phase shift on backscatter. OK corrected

14. P.6, l.175: the units of these ranges of values (degree?) should be provided. done

15. P.6, l.179: the assumption of negligible $\delta hv$ should be better justified. A few degrees for $\delta hv$ as suggested on l.175 are not necessarily negligible compared to the overall $\Psi dp$ values provided in Fig.10 for instance. As mentioned in the General Comments, the resulting uncertainty in $\Phi dp$ values may affect the behavior highlighted in Fig.10 and 11. Combined with possible radome attenuation... Yes we agree, but there is little possibility to go further... We have expended and moderated our comments of new Figs 12 and 13 in section 4.2 and in the conclusion.

16. P.6, l.182-183: why N = 10 and N = 4? How did you come up with these values? This is empirical.

17. P.7, l.191: same here, please justify these thresholds in Zh and hv. Actually for the $\Phi\_(dp )$ profile processing for the convective case (without ML interaction), we determine now the beginning and the end in range of rain cells undisturbed by clutter by using a rhohv threshold only (_hv $\geq$ 0.95) to be valid over a number of successive gates (10 gates for XPORT radar, i.e. a range extent of 342 m). We had to adapt these figures for the MOUC radar due to the well-known decrease of rhohv in the ML (_hv $\geq$ 0.80, and 2 successive gates, that is 480 m) and to consider the actual range of the first mountain gate for the determination of rM.

18. P.7, l.196: the black line in Fig.4 represents the instantaneous values of Zh, it would benice to figure the variability of the mountain return, to give the reader an idea about the noise of such echoes (and hence an idea about the uncertainty in the derived PIA estimates). We have added a lot of material in the revision about the stability and time variability of the dry-weather mountain returns of the various targtes. We did not modify Fig.4 however for which the considered mountain returns correspond only to those available for a given radial of the considered target.

19. P.8, l.218: please provide a reference for negligible attenuation in snow. done

20. P.8, l.228: "[7]" seems to be a literature reference, but there is no number in the references. Please update. corrected

21. P.8, l.245: the co-fluctuation between the two signals does not look that bad by eye... Maybe you could compute the correlation coefficient to have a quantitative criterion? The number of pairs of points is quite low for individual targets. The results for the ensemble of targets is displayed in new Fig.11 (old Fig 9).

22. P.8, l.242-248: the possible influence of beam broadening and radome attenuation (see l.401-406) could be first mentioned here. Yes, we have added a paragraph on the possible error sources at the end of section 3.2

23. P.9, l.277: change citations from numbers ([10] and [19]) to author's names... done

24. P.10, l.302 and Fig.8: I may be wrong, but I think there is an issue with the axis labels in Fig.8: PIA from polarimetry should be on the y axis while the PIA from MRT should be on the x-axis. Otherwise, there would be an underestimation from the polarimetric approach (slope > 1), not consistent with Fig.6 left. Please clarify. Very good point! But I confirm that the MRT PIA is on the y-axis and the polarimetry-derived PIA is on the x-axis. As explained in the revised manuscript, this effect is related to the different ranges of Kdp values considered in the DSD analyses and in the Kdp range profiles discretized at 34.2 m. We have checked the stability of the non-linear relation when considering 1-min DSD samples, leading to an extended Kdp range.

25. P.11, l.322-323: what can explain this variability in the ML depth? If this is due to different types of hydrometeors, is the scaling approach used here still relevant? We think this variability of the ML thickness (with high values between 2:30 and 3:00 UTC, visible on the bottom graph of Fig. 5) to be due to the arrival of the hotter air mass and some kind of atmospheric mixing. We preferred this altitude scaling to the display of absolute altitudes with respect to e.g. the reflectivity or the rhohv peak altitude since the ML thickness varies significantly and the curves of the various ML characteristic points (peaks and inflexion points of Zh and rhohv) evolve rather harmoniously during the ML rise.

26. P.11, l.342: could this less evident shift between peak in Zh and in hv be also due

to beam broadening? As the ML is going up in altitude, it is also going further away in the PPI used to extract the polarimetric radar variables... Maybe... However, in the more systematic study of the ML described in Khanal et al. 2019, the shift between the two peaks has been evidenced for MLs at the altitude range of the last part of the January 4th 2018 event. We don't think this point alters the conclusions made in this article.

27. P.12, l.379: why are $\delta$hv values expressed in dB? It was a mistake, corrected, thanks!

28. P.13, l.405-406: but the attenuation due to wet snow sticking on the radome is not necessarily directly proportional to the rain rate (it can accumulates...). The assumption of negligible radome attenuation during the ML scans should be better justified. As it could have significant impact on the estimated PIA values and hence on the behavior of the ratio PIA/$\Psi$dp in Fig.11. We have to mention that the radome of the MOUC radar is heated, so that snow may not accumulate that much over it. In addition snow/rain rates were low, so radome attenuation may be low in this case. But it is difficult to be sure, and this is certainly a major limitation of our current measurement configuration...

29. P.17, Table 1: the spectral width is not recorded? Yes for the MOUC radar, no for the XPORT radar at that time; but we didn't use Doppler data in this study

30. P.19, Fig.2: it would be better to use the same y axis scale between the 2 events, to ease the comparison. Not done! The required code was not available to the main author at the time of the revision (lockdown...)

31. P.20, Fig.3: the underlying images are too coarse in resolution. They should be improved. done

32. P.21, Fig.4: As expected, the phase measurements are contaminated by clutter earlier (i.e. closer to the radar) than reflectivity measurements. Hence the last (starting from the radar) reliable gate in $\Psi$dp may be closer to the radar than the last reliable

gate in Zh from which the PIA is estimated. Could this introduce a bias? The $\Phi\_dp$ processing (and the subsequent polarimetry-derived PIA estimation) is made on the range of gates non contaminated by close-range or mountain clutter. The MRT PIA includes "on-site" attenuation and attenuation "over" the mountain. So yes, this could introduce a bias, which is difficult to estimate, but that we think of limited magnitude in the considered examples. This is mentioned in the revision.

---

## Author Response (AR2)

P.7, l.208 and 217: the values of the scatter around the mean (either quantified as the standard deviation or the interquantile) is relevant relative to the PIA estimates (most of the times > 10 dB). I think this should be mentioned to make clear to the reader that such uncertainties (roughly 1 dBZ for XPORT and 2-3 dBZ for MOUC) on the mountain echoes are quite acceptable.

Actually, more than the accuracy, I believe the time variability of the dry-weather returns to define the "sensitivity" of the MRT (the weight that has to be put on the scale pan so that the needle starts to move). I don't find appropriate to include such a comment on the MRT PIA accuracy at this point of the article. We already discussed this point in the conclusion in some sentences that are slightly modified in the revision:

Old line 540 (no modification): The MRT sensitivity depends on the time variability of the dry-weather mountain returns.

Old lines 557-559 (modifications): The stability of the apparent reflectivity of the mountain targets was shown to be very good, an indication of a good radar calibration stability during the considered period. The time variability of the reference returns during dry-weather preceding or succeeding the rain events was also found to be very small with standard deviations in the range of [0.2 – 0.9 dBZ], enabling a MRT PIA sensitivity better than 1 dB.

Old line 610: The $\delta_{hv}$ effect is likely to be strong in the ML (up to 4°) and its relative importance may be quite high in our case study since the PIA range is significantly lower compared to the rain case study, with maximum PIAs of about 15 dB (note also that the sensitivity of the MRT is less than for the XPORT case study since the dry-weather variability of the mountain returns is higher with standard-deviations in the range [0.62-1.44]).

p.8, l. 249-250: "An anonymous reviewer..." please rephrase, as this is a valid question, independently of the reviewer.

We refer now to the article by Trömer et al. JAMC 2013:

Scattering simulations based on disdrometer data (Trömel et al. 2013) indicate that there may exist quite a large scatter with respect to such power-law models and an important influence of the considered hydrometeor temperature. From simulations based on radar data at various frequencies, the same authors quantify δhv values as high as 4° in the ML at X-Band and mention that strong δhv values may be associated with both large dry hailstones and wet hailstones.

p.9, l.258-259: it is nice to mention that these are empirical choices, but the authors should mostly explain if the outcome is sensitive to those empirical values or not... that was the meaning of my comment in the 1st round of reviews.

The sensitivity of these values was not extensively tested actually; the text was modified so to read as:

The raw $\psi_{dp}(r)$ values for which $\rho_{hv}(r)$ was less than 0.95 (empirical choice with limited impact in the [0.95-0.97] range) were set to missing values. In addition, we defined the beginning of the rainy range by determining the first series of 10 successive gates (again an empirical choice, corresponding to a range extent of 342 m) overpassing this threshold.

p.18, l.567: I may be wrong, but it is questionable to directly compare values of total differential

phase shift and values of diff phase shift on backscatter, as the former is integrated over a range while the latter is "local". And depending on the considered time period of the event, the total diff phase shift can be lower and in the order of 10-15 deg (e.g. Fig.6 and 7), in which case 4 deg is becoming more significant. So I would recommend the author to rephrase or modify this piece of text.

We agree that assuming deltahv to be negligible may lead to some conditional bias in the scatterplots displayed in old Fig. 6 and 7 (new figs 9 and 10), but as mentioned in the sentence preceding line 567, we are rather confident in the good ability of the regularization procedure to filter bumps in the raw psidp profiles, likely due to deltahv; and these bumps appear on a very limited number of profiles in our dataset. We acknowledge some lines after (old lines 610-612) that the deltahv impact may be greater for the MOUC case study due to both the limited PIA range and the (rather unknown) behaviour of deltahv in the ML

Therefore we have simply modified the text at l. 567 as:

[revised manuscript text omitted]